# Secure Domain Adaptation with Multiple Sources

**Serban Stan**                                                                 *sstan@usc.edu*
*Department of Computer Science*
*University of Southern California*

**Mohammad Rostami**                                                            *rostamim@usc.edu*
*Department of Computer Science*
*University of Southern California*

**Reviewed on OpenReview:** *https: // openreview. net/ forum? id= UDmH3HxxPp&noteId*

## Abstract

Multi-source unsupervised domain adaptation (MUDA) is a framework to address the challenge of annotated data scarcity in a target domain via transferring knowledge from multiple annotated source domains. When the source domains are distributed, data privacy and security can become significant concerns and protocols may limit data sharing, yet existing MUDA methods overlook these constraints. We develop an algorithm to address MUDA when source domain data cannot be shared with the target or across the source domains. Our method is based on aligning the distributions of source and target domains indirectly via estimating the source feature embeddings and predicting over a confidence based combination of domain specific model predictions. We provide theoretical analysis to support our approach and conduct empirical experiments to demonstrate that our algorithm is effective.

## 1 Introduction

Advances in deep learning have led to significant increase in the performance of machine learning (ML) algorithms in many applications (Russakovsky et al. (2015)). However, deep learning relies on access to large quantities of labeled data for end-to-end blind training. Even if a neural model is trained using annotated data, in settings where domain-shift (Torralba & Efros (2011)) exists between the training domain and deployment domain, neural models have been shown to suffer from sub-optimal performance. In such scenarios, the naive solution is to retrain the models on each domain from scratch. This operation however necessitates annotating large databases persistently, which has proven to be a time-consuming and expensive process Rostami et al. (2018). Unsupervised Domain Adaptation (UDA) (Long et al. (2016)) is a framework developed to address the challenge of domain-shift and permit model generalization between a *source domain* which has access to *labeled data* and a related *target domain* in which only *unannotated data* is accessible.

Another recent line of research considers Invariant Risk Minimization (IRM) (Arjovsky et al. (2019)), where model generalization on multiple domains is desirable (Ahuja et al. (2020)). The key difference compared to the UDA framework is that in UDA minimizing target error is preferred to maintaining joint high performance on both source and target. Thus, UDA methods operate in a more relaxed environment compared to IRM, and allow for better target generalization (Krueger et al. (2020); Kamath et al. (2021)).

An effective technique to address UDA is to map data points from a source and a target domain into a shared latent embedding space at which distributions are aligned Rostami (2021b). The latent embedding space is often modeled as the output-space of a deep encoder, trained to produce a shared representation for both domains. This outcome can be achieved using adversarial learning (Hoffman et al. (2018); Dou et al. (2019); Tzeng et al. (2017); Bousmalis et al. (2017)), where the distributions are matched indirectly through competing generator and discriminator networks to learn a domain-agnostic embedding. Alternatively, a probability metric can be selected and minimized to align the distributions directly in the embedding (Chen et al. (2019); Sun et al. (2017); Lee et al. (2019); Rostami & Galstyan (2020); Stan & Rostami (2021)).

Most existing UDA algorithms consider a single source domain for knowledge transfer. Recently, single-source unsupervised domain adaptation (SUDA) has been extended to multi-source unsupervised domain adaptation (MUDA), where several distinct sources of knowledge are available (Peng et al. (2019a); Zhao et al. (2020); Rostami et al. (2019); Lin et al. (2020); Guo et al. (2020); Tasar et al. (2020); Venkat et al. (2020a)). The goal in MUDA is to benefit from the collective information encoded in several distinct annotated source domains to improve model generalization on an unannotated target domain. Compared to SUDA, MUDA algorithms require leveraging data distribution discrepancies between pairs of source domains, as well as between the sources and the target. Thus, an assumption in most MUDA algorithms is that the annotated source datasets are centrally accessible. Such a premise however ignores privacy/security regulations or bandwidth limitations that may constrain the possibility of joint data access between source domains.

In practice, it is natural to assume source datasets are distributed amongst independent entities, and sharing data between them may constitute a privacy violation. For example, improving mobile keyboard predictions is performed by securely training models on independent computing nodes without centrally collecting user data (Yang et al. (2018)). Similarly, in medical image processing applications, data is often distributed amongst different medical institutions. Due to privacy regulations (Yan et al. (2021)) sharing data can be prohibited, and hence central access to data for all the source domains simultaneously becomes infeasible. MUDA algorithms can offer privacy between the sources and target by operating in a source-free regime (Ahmed et al. (2021)), i.e., during the adaptation process source samples are considered to be unavailable, and only the source trained model or source data statistics are assumed to be accessible. However, approaches operating under this premise require retraining if new source domains become available, or if a set of source domains becomes inaccessible. This downside leads to increased time and computational resource cost.

We relax the need for centralized processing of source data in MUDA while maintaining cross-domain privacy. Our approach is robust to accessibility changes for different source domains, allowing for relearning the target decision function without end-to-end retraining. We relax the need for direct access to source domain samples for adaptation by approximating the distribution of source embeddings. We perform source-free adaptation with respect to each source domain, and propose a confidence based pooling mechanism for target inference. In the present work, we: (i) Address the challenge of data privacy for MUDA by maintaining full privacy between pairs of source domains, and between the sources and the target. (ii) Propose an efficient distributed optimization process for MUDA to process each dataset locally while encoding high-level learned knowledge in a shared latent embedding space. (ii) Provide theoretical justification for our method by proving that our algorithm minimizes an upper-bound of the target error. We conduct extensive empirical experimental results on five standard MUDA benchmark datasets to demonstrate the effectiveness of our approach.

## 2 Related work

**Single-Source UDA:** Single source UDA aims to improve model generalization for an unlabeled target domain using only a single source domain with annotated data. SUDA has been studied extensively. A primary workflow employed in recent UDA works consists of training a deep neural network jointly on the labeled source domain and the unlabeled target domain to achieve distribution alignment between both domains in a latent embedding space. This goal has been achieved by employing generative adversarial networks (Goodfellow et al. (2014)) to encourage domain alignment (Hoffman et al. (2018); Dhouib et al. (2020); Luc et al. (2016); Tzeng et al. (2017); Sankaranarayanan et al. (2018)) as well as directly minimizing an appropriate distributional distance between the source and target embeddings (Long et al. (2015; 2017b); Morerio et al. (2018)). SUDA algorithms do not leverage inter-domain statistics in the presence of several source domains, and thus extending single-source UDA algorithms to a multi-source setting is nontrivial.

**Multi-Source UDA:** The MUDA setting is a recent extension of SUDA, where multiple streams of data are concomitantly leveraged for improved target domain generalization. Xu et al. (2018) minimize discrepancy between source and target domains by optimizing an adversarial loss. Peng et al. (2019a) adapt on multiple domains by aligning inter-domain statistics of the source domains in an embedding space. Guo et al. (2018) learn to combine domain specific predictions via meta-learning. Venkat et al. (2020a) use pseudo-labels to improve domain alignment. The increased amount of source data in MUDA is not necessarily an advantage over SUDA, as negative transfer between domains needs to be controlled during adaptation. Li et al. (2018)

exploit domain similarity to avoid negative transfer by leveraging model statistics in a shared embedding space. Zhu et al. (2019) achieve domain alignment by adapting deep networks at various levels of abstraction. Zhao et al. (2020) align target features against source trained features via optimal transport, then combine source domains proportionally to Wasserstein distance. Wen et al. (2020) use a discriminator to exclude data samples with negative generalization impact. Such approaches admit joint access to source domains during the training process, making them infeasible in settings where data privacy and security are of concern.

**Privacy in Domain Adaptation:** The importance of inter-domain privacy has been recognized and explored for single-source UDA, specifically in source-free adaptation. Note this framework is relevant in many important practical settings even for SUDA, where privacy regulations limit the possibility of sharing data (Peng et al. (2019b); Li et al. (2020); Liang et al. (2020a; 2021)). Several UDA approaches consider maintaining an approximatinon of the source domain for adaptation. Kurmi et al. (2021)) benefit from GANs to generate source-domain like samples during the adaptation phase. Yeh et al. (2021) align distributions via minimizing the KL-divergence in addition to a variational autoencoder reconstruction loss. Similar to our approach, Yang et al. (2022) model the source distribution and use clustering of target samples to assign the correct class is done by minimizing an VAE reconstruction error. Tian et al. (2022) approximate the latent source space during adaptation and use adversarial learning in their approach. Ding et al. (2022) also estimate the source distribution and choose specific anchors to guide the distribution learning process. Adaptation is done by optimizing class conditional maximum mean discrepancy between samples from the learnt approximation distributions and the target samples. These above works consider only a single source. In multi-source adaptation, the strategy for combining information across source domains is key in achieving competitive performance. Peng et al. (2019b) perform collaborative adaptation under privacy restrictions between source domains under the framework of federated learning. Ahmed et al. (2021) approach privacy-preserving MUDA via information maximization and pseudo-labeling. Dong et al. (2021) choose high confidence target samples as class anchors and pseudo-labels are then assigned according to the closest anchors. Unlike our approach, Ahmed et al. (2021); Dong et al. (2021) require simultaneous access to all source trained models during adaptation. This makes these method unsuitable for scenarios such as asynchronous federated learning (Peng et al. (2019b)), where communication between individual domains may be broken, or processing different source domains may be done at irregular time intervals (McMahan et al. (2016)). Our main adaptation tool is represented by optimal transport based optimization, followed by a confidence based pooling mechanism, making the adaptation process considerably more lightweight.

We address a more constrained yet practical setting, where privacy should be preserved both between pairs of source domains and with respect to the target. Our approach allows for efficient distributed optimization, not requiring end-to-end retraining if different source domains become inaccessible due to privacy obligations McMahan et al. (2016); Peng et al. (2019b), or more source domains become available after initial training has finished, allowing for accumulative learning from several domains. Our method builds on extending the idea of probability metric minimization, explored in UDA (Chen et al. (2019); Lee et al. (2019); Stan & Rostami (2022); Rostami (2021a; 2022)) to MUDA. The latent source and target features are represented via the output space of a neural encoder. Domain alignment implies a shared embedding space for these representations. To achieve this, a suitable distributional distance metric is chosen between these two sets of embeddings and minimized. In this work, we used the Sliced Wasserstein Distance (SWD) (Rabin et al. (2011); Bonneel et al. (2015)) for this purpose. SWD is a metric for approximating the optimal transport metric (Redko et al. (2019)). It is a suitable choice for UDA because: (i) it possesses non-vanishing gradients for two distributions with non-overlapping supports. As a result, it is a suitable objective function for deep learning gradient-based optimization techniques. (ii) It can be computed efficiently based on a closed-form solution using only empirical samples, drawn from the two probability distributions.

## 3 Problem formulation

Let $\mathcal{S}_1, \mathcal{S}_2 \ldots \mathcal{S}_n$ be the data distributions of $n$ annotated source domains and $\mathcal{T}$ be the data distribution of an unannotated target domain. Assume the source and target domains share the same feature space $\mathbb{R}^{W \times H \times C}$, where $W, H, C$ describe an image by width, height and number of channels respectively. We consider all domains having a common label-space $\mathcal{Y}$, but not necessarily sharing the same label distribution. For each source domain $k$, we observe the labeled samples $\{(\boldsymbol{x}_{k,1}^s, \boldsymbol{y}_{k,1}), \ldots, (\boldsymbol{x}_{k,n_k^s}^s, \boldsymbol{y}_{k,n_k^s})\}$, where $\boldsymbol{x}_k^s \sim \mathcal{S}_k$.

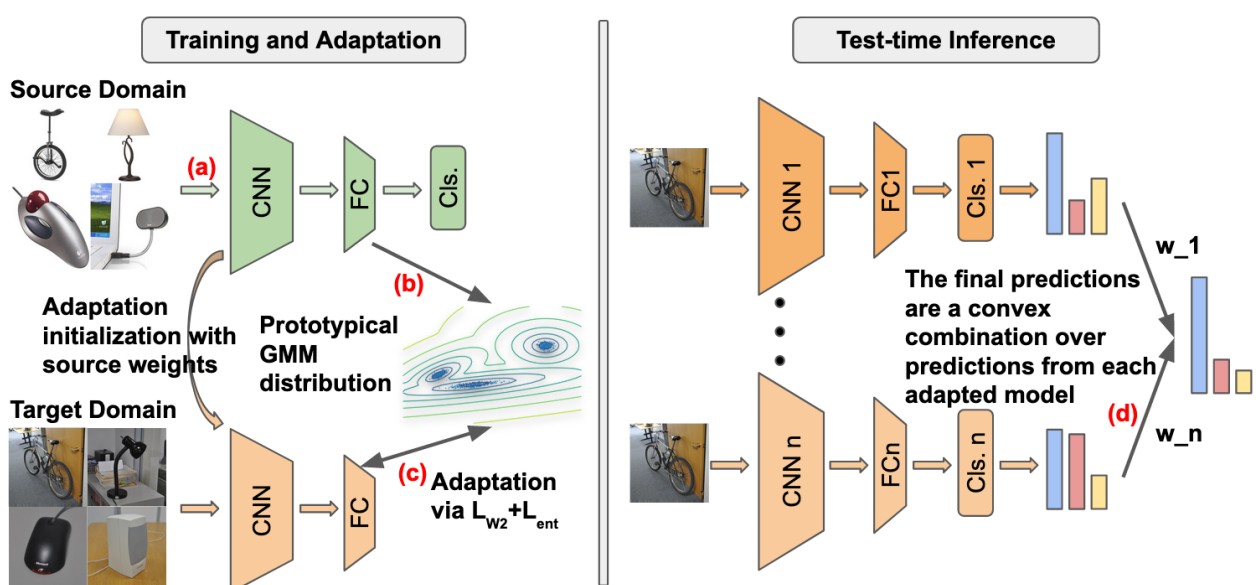

Figure 1: Block-diagram of our proposed approach: (a) source specific model training is done independently for each source domain (b) the distribution of latent embeddings of each source domain is estimated via a mixture of Gaussians, (c) for each source trained model, adaptation is performed by minimizing the distributional discrepancy between the learnt GMM distribution and the target encodings (d) the final target domain predictions are obtained via a learnt convex combinations of logits for each adapted model

We only observe unlabeled samples $\{\boldsymbol{x}_1^t, \ldots, \boldsymbol{x}_{n^t}^t\}$ from the target domain $\mathcal{T}$. The goal is to train a model $f_\theta : \mathbb{R}^{W \times H \times C} \to \mathbb{R}^{|\mathcal{Y}|}$ capable of inferring target labels, where $|\mathcal{Y}|$ is the number of inference classes. The first step in our approach is to independently train decision models for each source domain via empirical risk minimization (ERM) by minimizing the cross-entropy loss $\mathcal{L}_{ce}$: $\theta_k = \arg\min_\theta \frac{1}{n_k^s} \sum_{i=1}^{n_k^s} \mathcal{L}_{ce}(f_\theta(\boldsymbol{x}_{k,i}^s), \boldsymbol{y}_{k,i})$. Since under our considered setting the target and source domains share a common input and label space, these models can be directly used on the target to derive a naive solution. However, given the distributional discrepancy between source domains and target, e.g., real world images versus clip art, generalization performance will be poor. The goal of our MUDA approach is to benefit from the unannotated target dataset and the source-trained models in order to improve upon model generalization while avoiding negative transfer.

To this end, we decompose the model $f_\theta$ into a feature extractor $g_{\boldsymbol{u}}(\cdot) : \mathbb{R}^{W \times H \times C} \to \mathbb{R}^{d_Z}$ and a classifier subnetwork $h_{\boldsymbol{v}}(\cdot) : \mathbb{R}^{d_Z} \to \mathbb{R}^{|\mathcal{Y}|}$ with learnable parameters $\boldsymbol{u}$ and $\boldsymbol{v}$, such that $f(\cdot) = (h \circ g)(\cdot)$. We assume input data points are images of size $W \times H \times C$ and the latent embedding shape is of size $d_Z$. In a SUDA setting, we can improve generalization of each source-specific model on the target domain by aligning the distributions of the source and the target domain in the latent embedding space. Specifically, we can minimize a distributional discrepancy metric $D(\cdot, \cdot)$ across both domains, e.g., SWD loss, to update the learnable parameters: $\boldsymbol{u}_k^A = \arg\min_{\boldsymbol{u}} D(g_{\boldsymbol{u}}(\mathcal{S}_k), g_{\boldsymbol{u}}(\mathcal{T}))$. By aligning the distributions the source trained classifier $h_k$ will generalize well on the target domain $\mathcal{T}$. In the MUDA setting, the goal is improving upon SUDA by benefiting from the collective knowledge of the source domains to make predictions on the target. This can be done via a weighted average of predictions made by each of the domain-specific models, i.e., models with learnable parameters $\theta_k^A = (\boldsymbol{u}_k^A, \boldsymbol{v}_k)$. For a sample $\boldsymbol{x}_i^t$ in the target domain, the model prediction will be $\sum_{k=1}^n w_k f_{\theta_k^A}(X_i^t)$, where $w_k$ denotes a set of learnable weights associated with the source domains.

We note the above general approach requires simultaneous access to source and target data during adaptation. We relax this constraint and consider the more challenging setting of source-free adaptation, where we lose access to the source domains once source training finishes. To account for applications with sensitive data, e.g. medical domains, we also forbid interaction between source models during adaptation. Hence, the source distributions $S_k$ and their representations in the embedding space, i.e., $g(\mathcal{S}_k)$, will become inaccessible. To circumvent this challenge, we rely on intermediate distributional estimates of the source latent embeddings.

## 4 Proposed algorithm

Our proposed approach for MUDA with private data is visualized in Figure 1. We base our algorithm on two levels of hierarchies. First, we adapt each source-trained model while preserving privacy (left and middle subfigures). We then combine predictions of the source-specific models on the target domain according to their reliability (right subfigure). To tackle the challenge of data privacy, we approximate the distributions of the source domains in the embedding space as a multi-modal distribution and use these distributional estimates for domain alignment (Figure 1, left). We can benefit from these estimates because once source training is completed, the input embedding distribution should be mapped into a $|\mathcal{Y}|-$modal distribution to enable the classifier subnetwork to separate the classes. Note, each separated distributional mode encodes one of the classes (see Figure 1, left). To approximate these internal distributions we employ Gaussian Mixture Models (GMM), with learnable mean and covariance parameters $\mu_k, \Sigma_k$. Since we have access to labeled source data points, we can learn $\mu_k$ and $\Sigma_k$ in a supervised fashion. Let $\mathbb{1}_c(x)$ denotes the indicator function for $x = c$, then the maximum likelihood estimates for the GMM parameters would be:

$$\mu_{k,c} = \frac{\sum_{i=1}^{n_k^s} \mathbb{1}_c(\boldsymbol{y}_{k,i}) g_{\boldsymbol{u}_k}(\boldsymbol{x}_{k,i}^s)}{\sum_{i=1}^{n_k^s} \mathbb{1}_c(\boldsymbol{y}_{k,i})}, \quad \Sigma_{k,c} = \frac{\sum_{i=1}^{n_k^s} \mathbb{1}_c(\boldsymbol{y}_{k,i})(g_{\boldsymbol{u}_k}(\boldsymbol{x}_{k,i}^s) - \mu_{k,c})(g_{\boldsymbol{u}_k}(\boldsymbol{x}_{k,i}^s) - \mu_{k,c})^T}{\sum_{i=1}^{n_k^s} \mathbb{1}_c(\boldsymbol{y}_{k,i})} \tag{1}$$

Learning $\mu_k$ and $\Sigma_k$ for each domain $k$ enables us to sample class conditionally from the GMM distributional estimates and approximate the distribution $g(\mathcal{S}_k)$ in the absence of the source dataset.

We adapt the source-trained model by aligning the target distribution $\mathcal{T}$ and the GMM distribution in the embedding space. To preserve privacy, for each source domain $k$ we generate intermediate pseudo-domains $A_k$ with pseudo-samples $\{\boldsymbol{z}_{k,1}^a, \ldots, \boldsymbol{z}_{k,n_k^a}^a\}$ by drawing random samples from the estimated GMM distribution. The pseudo-domain is used as an approximation of the corresponding source embeddings. To align the two distribution, a suitable distance metric $D(\cdot, \cdot)$ needs to be used. We rely on the SWD due to its mentioned appealing properties. The SWD acts as an estimate for the Wasserstein Distance (WD) between two distributions (Rabin et al. (2011)), by aggregating the tractable $1-$dimensional WD over $L$ projections onto the unit hypersphere. In the context of our algorithm, the SWD discrepancy measure becomes:

$$D(g(\mathcal{T}), A_k) = \frac{1}{L} \sum_{l=1}^{L} |\langle g(x_{i_l}^t), \phi_l \rangle - \langle z_{k,j_l}^a, \phi_l \rangle|^2 \tag{2}$$

where $\phi_l$ is a projection direction, and $i_l, j_l$ are indices corresponding to the sorted projections. While the source and target domains share the same label space $\mathcal{Y}$ they do not necessarily share the same distribution of labels. Since the prior probabilities on classes are not known in the target domain, minimizing the SWD at the batch level may lead to incorrectly clustering samples from different classes together, depending on the discrepancy between the label distributions. To address this challenge, we take advantage of the conditional entropy loss (Grandvalet & Bengio (2004)) as a regularization term based on information maximization. The conditional entropy acts as a soft clustering objective that ensures aligning target samples to the wrong class via SWD will be penalized. We follow the approximation presented in Eq. 6 in Grandvalet & Bengio (2004):

$$\mathcal{L}_{ent}(f_\theta(\mathcal{T})) = \frac{1}{n^t} \sum_{i=1}^{n^t} \mathcal{L}_{ce}(f_\theta(x_i^t), f_\theta(x_i^t)) \tag{3}$$

To guarantee this added loss term influences the latent representations produced by the feature extractor, the classifier is frozen during model adaptation. Our final combined adaptation loss is descrbied as:

$$D(g(\mathcal{T}), A) + \gamma \mathcal{L}_{ent}(f_\theta(\mathcal{T})) \tag{4}$$

for a regularizer $\gamma$. Once the source-specific adaptation is completed across all domains, the final model predictions on the target domain are obtained by combining probabilistic predictions returned by each of the $n$ domain-specific models. The mixing weights are chosen as a convex vector $\boldsymbol{w} = (w_1 \ldots w_n)$, i.e., $w_i > 0$ and $\sum_i w_i = 1$, with final predictions taking the form $\sum_{i=1}^{k} w_i f_{\theta_i}$. The choice of $w$ is critical, as assigning large weights to a model which does not generalize well will harm inference power. We utilize the source model *prediction confidence* on the target domain as a proxy for generalization performance. We have provided empirical evidence for this choice in Section 6. We thus set a confidence threshold $\lambda$ and assign $w_k$:

$$\tilde{w}_k \sim \sum_{i=1}^{n^t} \mathbb{1}(\max \tilde{f}_{\theta_k}(\boldsymbol{x}_i^t) > \lambda), \quad w_k = \tilde{w}_k / \sum \tilde{w}_k, \tag{5}$$

where $\tilde{f}(\cdot)$ denotes the model ouput just prior to the final SoftMax layer which correspond to a probability.

Note the only cross-domain information transfer in our framework is communicating the latent means and covariance matrices of the estimated GMMs plus the domain-specific model weights which provide a warm start for adaptation. Data samples are never shared between any two domains during pretraining and adaptation. As a result, our approach preserves data privacy for scenarios at which the source datasets are distributed across several entities. Additionally, the adaptation process for each source domain is performed independently. As a result, our approach can be used to incorporate new source domains as they become available over time without requiring end-to-end retraining from scratch. We will only require to update the normalized mixing weights via Equation 5, which takes negligible runtime compared to model training. Our proposed privacy preserving approach, named Secure MUDA (SMUDA), is presented in Algorithm 1 .

---

**Algorithm 1** Secure Multi-source Unsupervised Domain Adaptation (SMUDA)

> **procedure** SMUDA($\mathcal{S}_1 \ldots \mathcal{S}_n, \mathcal{T}, L, \gamma$)
>     **for** $k \leftarrow 1$ to $n$ **do**
>         $\mu_k, \Sigma_k, \theta_k = Train(\mathcal{S}_k)$
>         Generate $A_k$ based on $\mu_k, \Sigma_k$
>         Compute $w_k$ (Eq. 5)
>         $\theta_k^A = Adapt(\theta_k, A_k, \mathcal{T}, L, \gamma)$
>     **return** $w_1 \ldots w_n, \theta_1^A \ldots \theta_n^A$
> **procedure** TRAIN($\mathcal{S}_i$)
>     Learn $\theta_k = (u_k, v_k)$ by min. $\mathcal{L}_{CE}(f_{\theta_k}(\mathcal{S}_k), \cdot)$
>     Learn parameters $\mu_k, \Sigma_k$ (Eq. 1)
>     **return** $\mu_k, \Sigma_k, \theta_k$
> **procedure** ADAPT($\theta_k, A_k, \mathcal{T}, L, \gamma$)
>     Initialize network with weights $\theta_k$
>     $\theta_k^A = \arg\min_\theta D(g_u(\mathcal{T}), A_k) + \gamma \mathcal{L}_{ent}(f_\theta(\mathcal{T}))$
> (Eq. 4)
>     **return** $\theta_k^A$

---

## 5 Theoretical analysis

We provide an analysis to demonstrate that our algorithm minimizes an upper bound for the target domain error. We adopt the framework developed by Redko & Sebban (2017) for *single source UDA using Wasserstein distance* to provide a theoretical justification for the algorithm we proposed. Our analysis is performed in the latent embedding space. Let $\mathcal{H}$ represent the hypothesis space of all classifier subnetworks. Let $h_k(\cdot)$ denote the model learnt by each domain-specific model. We also set $e_{\mathcal{D}}(\cdot)$, where $\mathcal{D} \in \{\mathcal{S}_1 \ldots \mathcal{S}_n, \mathcal{T}\}$, to be the true expected error returned by some model $h(\cdot) \in \mathcal{H}$ in the hypothesis space on the domain $\mathcal{D}$. Additionally, let $\hat{\mu}_{\mathcal{S}_k} = \frac{1}{n_k^s} \sum_{i=1}^{n_k^s} f(g(\boldsymbol{x}_{k,i}^s))$, $\hat{\mu}_{\mathcal{P}_k} = \frac{1}{n_k^a} \sum_{i=1}^{n_k^a} \boldsymbol{x}_{k,i}^a$, and $\hat{\mu}_{\mathcal{T}} = \frac{1}{n^t} \sum_{i=1}^{n_k^s} f(g(\boldsymbol{x}_i^t))$ denote the empirical distributions that are built using the samples for the source domain, the intermediate pseudo-domain, and the target domain in the latent space. Then the following theorem holds for the MUDA setting:

**Theorem 1.** *Consider Algorithm 1 for MUDA under the explained conditions, then the following holds*

$$e_{\mathcal{T}}(h) \leq \sum_{k=1}^{n} w_k \left( e_{\mathcal{S}_k}(h_k) + D(\hat{\mu}_{\mathcal{T}}, \hat{\mu}_{\mathcal{P}_k}) + D(\hat{\mu}_{\mathcal{P}_k}, \hat{\mu}_{\mathcal{S}_k}) + \sqrt{\left(2\log(\frac{1}{\xi})/\zeta\right)}\left(\sqrt{\frac{1}{N_k}} + \sqrt{\frac{1}{M}}\right) + e_{\mathcal{C}_k}(h_k^*) \right) \tag{6}$$

*where $\mathcal{C}_k$ is the combined error loss with respect to domain $k$, and $h_k^*$ is the optimal model with respect to this loss when a shared model is trained jointly on annotated datasets from all domains simultaneously.*

***Proof:*** the complete proof is included in the Appendix.

We note the target domain error is upper bounded by the convex combination of the domain-specific adaptation errors. Algorithm 1 minimizes the right-hand side of Equation 6 as follows: for each source domain, the source expected error is minimized by training the models using ERM. The second term is minimized by closing the distributional gap between the intermediate pseudo-domain and the target domain in the latent space. The third term corresponds to how well the GMM distribution approximates the latent source samples. Our algorithm does not directly minimize this term, however if the model forms a multi-modal distribution in the source embedding space, necessary for good performance, this term will be small. The second to last term is dependent on the number of available samples in the adaptation problem, and becomes negligible when sufficient samples are accessible. The final term measures the difficulty of the optimization, and is dependent only on the structure of the data. For related domains, this term will also be negligible.

## 6  Experimental validation

**Datasets** We validate on five datasets: *Office-31*, *Office-Home*, *Office-Caltech*, *Image-Clef* and *DomainNet*.

**Office-31** (Saenko et al. (2010)) is a dataset with 31 classes consisting of 4110 images from an office environment pertaining to three domains: Amazon, Webcam and DSLR. Domains differ in image quality, background, number of samples and class distributions. **Office-Caltech** (Gong et al. (2012)) contains 2533 from 10 classes of office related images from four domains: Amazon, Webcam, DSLR, Caltech. **Office-Home** (Venkateswara et al. (2017)) contains 65 classes and 30475 from four different domains: Art (stylized images), Clipart (clip art sketches), Product (images with no background) and Real-World (realistic images), making it more challenging that *Office* datasets. **Image-Clef** (Long et al. (2017a)) contains 1800 images under 12 generic categories sourced from three domains: Caltech, Imagenet and Pascal. **DomainNet** (Peng et al. (2019a)) is a larger, more recent dataset containing 586,575 images from 345 general classes, with different class distributions for each of its domains: Quickdraw, Clipart, Painting, Infograph, Sketch, Real.

**Preprocessing & Network structure:** we follow the literature for fair comparison. For each domain we re-scale images to a standard size of $(224, 224, 3)$. We use a ResNet50 (He et al. (2016)) network as a backbone for the feature extractor, followed by four fully connected layers. The network classification head consists of a linear layer, and source-training is performed using cross-entropy loss. The ResNet layers of the feature extractor are frozen during adaptation. We report classification accuracy, averaged across five runs. As hardware we used a and NVIDIA Titan Xp GPU. Our code is provided as part of the supplementary material and available online at https://github.com/serbanstan/secure-muda.

To test the effectiveness of our privacy preserving approach for MUDA, we compare our method against state-of-the art SUDA and MUDA approaches. Benchmarks for single best (SB), source combined (SC) and multi source (MS) performance are reported based on DAN (Long et al. (2015)), D-CORAL (Sun & Saenko (2016)), RevGrad (Ganin & Lempitsky (2015)). We include most existing MUDA algorithms: DCTN (Xu et al. (2018)), FADA (Peng et al. (2019b)), MFSAN (Zhu et al. (2019)), MDDA (Zhao et al. (2020)), SimpAl (Venkat et al. (2020b)), JAN (Long et al. (2017b)), MEDA (Wang et al. (2018)), MCD (Saito et al. (2018)), M3SDA (Peng et al. (2019a)), MDAN (Zhao et al. (2018)), MDMN (Li et al. (2018)), DARN (Wen et al. (2020)), DECISION (Ahmed et al. (2021)). Note that we maintain full domain privacy throughout training and adaptation. Hence, most of the above works should be considered as **upperbounds** in performance, as they address a more relaxed problem, by allowing joint and persistent access to source data. While Ahmed et al. (2021) also performs source-free adaptation, they benefit from jointly accessing the source trained models during adaptation, while our method only assumes joint access when pooling predictions. Despite this additional constraint, results prove our algorithm is competitive and at times outperforming the aforementioned methods. We next present quantitative and qualitative analysis of our work.

### 6.1  Performance Results

Table 1 presents our main results. For **Office-31**, we observe state-of-the-art performance (SOTA) on the $\to D, \to A$ tasks and near SOTA performance on the remaining task. Note that the domains *DSLR* and *Webcam* share similar distributions, as exemplified through the Source-Only results, and for these domains obtaining a good adaptation performance involves minimizing negative transfer, which our method

successfully achieves. In the case of **Image-clef**, we obtain SOTA performance on all tasks, even though the methods we compare against are not source-free. On the **Office-caltech** dataset, we obtain SOTA performance on the $\rightarrow A$ task, with close to SOTA performance on the three other tasks. The domains of the **Office-home** dataset have larger domain gaps with more classes compared to the three previous datasets. Our approach obtains near SOTA performance on the $\rightarrow P$ and $\rightarrow R$ tasks and competitive performance on the remaining tasks. Finally, the **DomainNet** dataset contains a much larger number of classes and variation in class distributions compared to the other datasets, making it the most challenging considered task in our experiments. Even so, we are able to obtain SOTA performance on three of the six tasks with competitive results on the other three. We reiterate most other MUDA algorithms serve as upper-bounds to our work, as they either access source data directly, simultaneously use models from all sources for adaptation, or both. Results across all tasks demonstrate that not only are we able to compare favorably against these methods while preserving data privacy, but we also set new SOTA on several tasks.

|     | Method | →D | →W | →A | Avg. |
| --- | --- | --- | --- | --- | --- |
| SB | Source Only | 99.3 | 96.7 | 62.5 | 86.2 |
| | DAN | 99.7 | 98.0 | 65.3 | 87.7 |
| | D-CORAL | 99.7 | 98.0 | 65.3 | 87.7 |
| | RevGrad | 99.1 | 96.9 | 66.2 | 87.5 |
| SC | DAN | 99.6 | 97.8 | 67.6 | 88.3 |
| | D-CORAL | 99.3 | 98.0 | 67.1 | 88.1 |
| | RevGrad | 99.7 | 98.1 | 67.6 | 88.5 |
| MS | MDDA | 99.2 | 97.1 | 56.2 | 84.2 |
| | DCTN | 99.3 | 98.2 | 64.2 | 87.2 |
| | MFSAN | 99.5 | 98.5 | 72.7 | 90.2 |
| | SImpAl | 99.2 | 97.4 | 70.6 | 89.0 |
| | DECISION* | 99.6 | 98.4 | **75.4** | 91.1 |
| | **SMUDA*+** | **99.8** | **98.5** | **75.4** | 91.2 |

(a) Office-31

|     | Method | →P | →C | →I | Avg. |
| --- | --- | --- | --- | --- | --- |
| SB | Source Only | 74.8 | 91.5 | 83.9 | 83.4 |
| | DAN | 75.0 | 93.3 | 86.2 | 84.8 |
| | D-CORAL | 76.9 | 93.6 | 88.5 | 86.3 |
| | RevGrad | 75.0 | 96.2 | 87.0 | 86.1 |
| SC | DAN | 77.6 | 93.3 | 92.2 | 87.7 |
| | D-CORAL | 77.1 | 93.6 | 91.7 | 87.5 |
| | RevGrad | 77.9 | 93.7 | 91.8 | 87.8 |
| MS | DCTN | 75.0 | 95.7 | 90.3 | 87.0 |
| | MFSAN | 79.1 | 95.4 | 93.6 | 89.4 |
| | SImpAl | 77.5 | 93.3 | 91.0 | 87.3 |
| | **SMUDA*+** | **79.4** | **96.9** | **93.9** | 90.1 |

(b) Image-clef

|     | Method | →W | →D | →C | → A | Avg. |
| --- | --- | --- | --- | --- | --- | --- |
| SB | Source Only | 99.0 | 98.3 | 87.8 | 86.1 | 92.8 |
| | DAN | 99.3 | 98.2 | 89.7 | 94.8 | 95.5 |
| | JAN | 99.4 | 99.4 | 91.2 | 91.8 | 95.5 |
| MS | DAN | 99.5 | 99.1 | 89.2 | 91.6 | 94.8 |
| | DCTN | 99.4 | 99.0 | 90.2 | 91.6 | 94.8 |
| | MEDA | 99.3 | 99.2 | 91.4 | 92.9 | 95.7 |
| | MCD | 99.5 | 99.1 | 91.5 | 92.1 | 95.6 |
| | M³SDA | 99.4 | 99.2 | 91.5 | 94.1 | 96.1 |
| | SImpAl | 99.3 | 99.8 | 92.2 | 95.3 | 96.7 |
| | FADA*+ | 88.1 | 87.1 | 88.7 | 84.2 | 87.1 |
| | DECISION* | **99.6** | **100** | **95.9** | **95.9** | 98.0 |
| | **SMUDA*+** | 99.3 | 97.6 | 93.9 | **95.9** | 96.6 |

(c) Office-caltech

|     | Method | →A | →C | →P | → R | Avg. |
| --- | --- | --- | --- | --- | --- | --- |
| SB | Source Only | 65.3 | 49.6 | 79.7 | 75.4 | 67.5 |
| | DAN | 68.2 | 56.5 | 80.3 | 75.9 | 70.2 |
| | D-CORAL | 67.0 | 53.6 | 80.3 | 76.3 | 69.3 |
| | RevGrad | 67.9 | 55.9 | 80.4 | 75.8 | 70.0 |
| SC | DAN | 68.5 | 59.4 | 79.0 | 82.5 | 72.4 |
| | D-CORAL | 68.1 | 58.6 | 79.5 | 82.7 | 72.2 |
| | RevGrad | 68.4 | 59.1 | 79.5 | 82.7 | 72.4 |
| MS | MFSAN | 72.1 | 62.0 | 80.3 | 81.8 | 74.1 |
| | M³SDA | 64.1 | 62.8 | 76.2 | 78.6 | 70.4 |
| | SImpAl | 70.8 | 56.3 | 80.2 | 81.5 | 72.2 |
| | MDAN | 68.1 | 67.0 | 81.0 | 82.8 | 74.8 |
| | MDMN | 68.7 | 67.6 | 81.4 | 83.3 | 75.3 |
| | DARN | 70.0 | 68.4 | 82.8 | 83.9 | 76.2 |
| | DECISION* | **74.5** | 59.4 | **84.4** | **83.6** | 75.5 |
| | **SMUDA*+** | 69.1 | **61.5** | 83.5 | 83.4 | 74.4 |

(d) Office-home

|     | Method | → Q | → C | → P | → I | → S | → R | Avg. |
| --- | --- | --- | --- | --- | --- | --- | --- | --- |
| SB | Source Only | 11.8 | 39.6 | 33.9 | 8.2 | 23.1 | 41.6 | 26.4 |
| | DAN | 16.2 | 39.1 | 33.3 | 11.4 | 29.7 | 42.1 | 28.6 |
| | JAN | 14.3 | 35.3 | 32.5 | 9.1 | 25.7 | 43.1 | 26.7 |
| | ADDA | 14.9 | 39.5 | 29.1 | 14.5 | 30.7 | 41.9 | 28.4 |
| | MCD | 3.8 | 42.6 | 42.6 | 19.6 | 33.8 | 50.5 | 32.2 |
| SC | Source Only | 13.3 | 47.6 | 38.1 | 13.0 | 33.7 | 51.9 | 32.9 |
| | DAN | 15.3 | 45.4 | 36.2 | 12.8 | 34.0 | 48.6 | 32.1 |
| | JAN | 12.1 | 40.9 | 35.4 | 11.1 | 32.3 | 45.8 | 29.6 |
| | ADDA | 14.7 | 47.5 | 36.7 | 11.4 | 33.5 | 49.1 | 32.2 |
| | MCD | 7.6 | 54.3 | 45.7 | 22.1 | 43.5 | 58.4 | 38.5 |
| MS | DCTN | 7.2 | 48.6 | 48.8 | 23.5 | 47.3 | 53.5 | 38.2 |
| | M³SDA-$\beta$ | 6.3 | 58.6 | 52.3 | 26.0 | 49.5 | 62.7 | 42.6 |
| | FADA*+ | 7.9 | 45.3 | 38.9 | 16.3 | 26.8 | 46.7 | 30.3 |
| | DECISION* | **18.9** | 61.5 | **54.6** | 21.6 | **51** | 67.5 | 45.9 |
| | **SMUDA*+** | 14.6 | **62.4** | 53.6 | **24.4** | 49.9 | **68.3** | 45.5 |

(e) DomainNet

Table 1: Results on five benchmark datasets. Single best (SB) represents the best performance with respect to any source, source combined (SC) represents performance obtained by pooling the source data together from different domains, and multi source (MS) represents methods performing multi source adaptation. * indicates source-free adaptation, guaranteeing privacy between sources and the target. + indicates privacy between source models. Results in bold correspond to the highest accuracy amongst the source-free approaches.

## 6.2 Ablative Experiments and Empirical Analysis

We perform ablative experiments by investigating the effect of each loss term in Eq. 4 on performance, and present results in Table 2. We observe combining the two terms yields improved performance for all datasets besides *Office-caltech*, where the difference is negligible. On the other hand, minimizing the SWD is more impactful on the *Image-clef* and *Office-home* datasets. The conditional entropy contributes more for *Office-caltech* and some *Office-31* tasks. Our insight is conditional entropy is more impactful when the source trained models have higher initial performance on the target (e.g., $\rightarrow D, \rightarrow W$ on *Office-31*), while the

| Method | →D | →W | →A | Avg. |
|---|---|---|---|---|
| SWD only | 92.2 | 94.1 | 73.1 | 86.4 |
| $\mathcal{L}_{ent}$ only | 99.8 | 98.1 | 66.2 | 88.1 |
| SMUDA | 99.8 | 98.5 | 75.4 | 91.2 |

(a) Office-31

| Method | →W | →D | →C | → A | Avg. |
|---|---|---|---|---|---|
| SWD only | 98.1 | 97.8 | 92.1 | 95.5 | 95.9 |
| $\mathcal{L}_{ent}$ only | 99.4 | 97.7 | 94 | 96 | 96.8 |
| SMUDA | 99.3 | 97.6 | 93.9 | 95.9 | 96.6 |

(b) Office-caltech

| Method | →P | →C | →I | Avg. |
|---|---|---|---|---|
| SWD only | 79.3 | 96.5 | 94.2 | 90 |
| $\mathcal{L}_{ent}$ only | 78.5 | 96 | 91.7 | 88.7 |
| SMUDA | 79.4 | 96.9 | 93.9 | 90.1 |

(c) Image-clef

| Method | →A | →C | →P | → R | Avg. |
|---|---|---|---|---|---|
| SWD only | 66.6 | 59.1 | 80.9 | 82.2 | 72.2 |
| $\mathcal{L}_{ent}$ only | 64.5 | 49.4 | 77.8 | 72.2 | 66 |
| SMUDA | 69.1 | 61.5 | 83.5 | 83.4 | 74.4 |

(d) Office-home

Table 2: Results when only the SWD objective, the entropy objective or both (SMUDA) are used.

SWD term is more beneficial when there is a larger discrepancy between the source domains and the target domain (e.g., → A on *Office-31*). Experiments conclude using both terms further improves performance.

Preserving privacy when performing adaptation limits information access between domains. It is then expected that privacy preserving methods are at a disadvantage compared to methods that do not impose any privacy restrictions. We show that in the case of our method, this observed disadvantage is negligible. To study the effect of preserving privacy on UDA performance, we perform experiments where source data is shared either between sources or with the target domain. We consider three primary scenarios for sharing source data: (i) **SW**D loss is computed using the source domain latent features (SW); (ii) **S**ource domains' data is **C**ombined into a single source (SC); **S**upervised **S**ource loss is computed for joint UDA (SS). We report results for natural combinations of these approaches in Table 3. We observe SMUDA performs similarly to SW. Joint adaptation (SS), source-combined performance (SC), or a combination of the two offer improved performance on all datasets. The improvements from sacrificing privacy are however negligible compared to SMUDA. We conclude our source domain approximation using GMMs captures sufficient source information under the considered test cases. This leads our adaptation approach to achieve comparable performance to settings where privacy is not enforced, with the added benefit of not sharing data between domains.

| Method | →D | →W | →A | Avg. |
|---|---|---|---|---|
| SW | 99.5 | 98.5 | 75.5 | 91.2 |
| SC | 98.1 | 96.8 | 76 | 90.3 |
| SS | **99.8** | **98.7** | **76.3** | **91.6** |
| SW+SS | **99.8** | 98.6 | 75.7 | 91.3 |
| SC+SS | 98.4 | 96.9 | 76.1 | 90.5 |
| SC+SW+SS | 99.0 | 97.7 | 76.1 | 90.9 |
| SMUDA | **99.8** | 98.5 | 75.4 | 91.2 |

(a) Office-31

| Method | →W | →D | →C | → A | Avg. |
|---|---|---|---|---|---|
| SW | 99.4 | 96.9 | 93.9 | 95.9 | 96.5 |
| SC | **99.7** | 96.8 | 94.1 | 96 | 96.6 |
| SS | 99.6 | 97.2 | 94.1 | 95.9 | **96.7** |
| SW+SS | **99.7** | 97.4 | 94.1 | 96 | 96.8 |
| SC+SS | **99.7** | 96.8 | **94.2** | **96.2** | **96.7** |
| SC+SW+SS | 99.6 | 97.2 | 93.3 | 95.9 | 96.5 |
| SMUDA | 99.3 | **97.6** | 93.9 | 95.9 | 96.6 |

(b) Office-caltech

| Method | →P | →C | →I | Avg. |
|---|---|---|---|---|
| SW | 79.5 | 95.2 | 91.3 | 88.6 |
| SC | 79.8 | 96.6 | **94.2** | **90.2** |
| SS | 79.4 | 95.6 | 91.6 | 88.9 |
| SW+SS | 79.4 | 95.6 | 91.8 | 88.9 |
| SC+SS | **79.9** | 96.6 | 93.1 | 89.8 |
| SC+SW+SS | 79.5 | 95.2 | 91.3 | 88.6 |
| SMUDA | 79.4 | **96.9** | 93.9 | 90.1 |

(c) Image-clef

| Method | →A | →C | →P | → R | Avg. |
|---|---|---|---|---|---|
| SW | 68.9 | 60.8 | 83.3 | 83.4 | 74.1 |
| SC | 69.6 | 62.9 | 85.3 | 84.7 | 75.6 |
| SS | 68.9 | 61.1 | 84 | 83.6 | 74.4 |
| SW+SS | 68.5 | 61 | 83.8 | 83.8 | 74.3 |
| SC+SS | **69.4** | **63** | **85.5** | **85** | **75.7** |
| SC+SW+SS | 68.8 | 62.7 | 85.1 | 84.5 | 75.3 |
| SMUDA | 69.2 | 61.1 | 83.2 | 83.5 | 74.3 |

(d) Office-home

Table 3: Results comparing SMUDA to non-private variants.

To compare our method against ensemble models of existing single-source source-free UDA (SFUDA) methods, we performed experiments on the Office-31 dataset. We compare against five recent SFUDA approaches in Table 4, and the ensemble of these methods in Table 5. We observe that although in the SFUDA setting,

despite being competitive, our method trails some of the methods, it outperforms the methods in MUDA. We conclude that our method alleviates the effect of negative transfer successfully and indeed can boost performance of a weaker single-source performance. We also note that we likely can improve the SFUDA performance for our method if we benefit from better probability metrics or model regularization.

| Method | A→D | W→D | A→W | D→W | D→A | W→A | Avg. |
|---|---|---|---|---|---|---|---|
| USFDA Kundu et al. (2020) | 64.5 | 96 | 71 | 93.3 | 62.8 | 63.6 | 75.2 |
| SHOT Liang et al. (2020b) | 94.0 | 99.9 | 90.1 | 98.4 | **74.7** | 74.3 | 88.6 |
| AFN Xu et al. (2019) | 90.7 | 99.8 | 90.1 | 98.6 | 73.0 | 70.2 | 87.1 |
| MDD Zhang et al. (2019) | 93.5 | **100** | 94.5 | 98.4 | 74.6 | 72.2 | 88.9 |
| GVB-GD Cui et al. (2020) | **95.0** | **100** | **94.8** | **98.7** | 73.4 | 73.7 | **89.3** |
| **SMUDA (ours)** | 92.7 | 99.8 | 87.9 | 98.5 | 72.1 | **75.4** | 87.7 |

Table 4: Single source results

| Method | →D | →W | →A | Avg. |
|---|---|---|---|---|
| USFDA Kundu et al. (2020) | 96.0 | 93.1 | 65.5 | 84.9 |
| SHOT-ens Ahmed et al. (2021) | 97.8 | 94.9 | 75.0 | 89.3 |
| AFN Xu et al. (2019) | 98.4 | 96.4 | 71.3 | 88.7 |
| MDD Zhang et al. (2019) | 95.4 | 99.3 | 74.1 | 89.6 |
| GVB-GD Cui et al. (2020) | 97.2 | 95.6 | 74.9 | 89.2 |
| **SMUDA-Uniform** | 94.2 | 74.8 | 75.4 | 86.4 |
| **SMUDA (ours)** | **99.8** | **98.5** | **75.4** | **91.2** |

Table 5: Uniformly combined predictions

We study the effect of hyper-parameters on SMUDA performance. We first empirically validate our approach for computing the mixing parameters $w_k$. We consider four scenarios for combining model predictions: (i) Eq. 5, (ii) setting weights proportional to SWD between the intermediate and the target domains (a cross-domain measure of distributional similarity), (iii) using a uniform average, and (iv) assigning all mixing weight to the model with best target performance. Average performance for tasks from four of the datasets are reported in Table 6. We observe our choice leads to maximum performance. Single best performance is able to slightly outperform on one dataset, however suffers on tasks where significant pairwise domain gap exists. This is expected, as using several domains is beneficial when they complement each other in terms of available information. Assigning weights proportional to $D(g(\mathcal{T}), A_k)$ may seem intuitive, given that similarity between pseudo-datasets and target latent features indicates better classifier generalization. However, this method performs better only compared to uniform averaging. We conclude model reliability is a superior criterion of combining predictions. Uniform averaging leads to decreased generalization on the target domain as it treats all domains equally. As a result, models with the least generalization ability on the target domain harm collective performance.

| Dataset | High confidence | W2 | Uniform | Single Best |
|---|---|---|---|---|
| Office-31 | 91.2 | 88.6 | 85.1 | 91.2 |
| Image-clef | 90.1 | 89.6 | 89.8 | 89.6 |
| Office-caltech | 96.6 | 96.6 | 96.6 | 97 |
| Office-home | 74.4 | 74.2 | 74.2 | 72.8 |
| Total avg. | **88.1** | 87.2 | 86.4 | 87.6 |

Table 6: Analytic experiments to study four strategies for combining the individual model predictions. Mixing based on model reliability proves superior to other popular approaches.

We additionally study the effect of the SWD projection hyper-parameter. SWD utilizes $L$ random projections, as detailed in Equation 2. While a large $L$ leads to a tighter approximation of the optimal transport metric, it also incurs a computational resource penalty. We investigate whether there is a range of $L$ values offering sufficient adaptation performance, and analyze the impact of this parameter using the *Office-31* dataset. In Figure 2 we reported performance results for $L \in \{1, 10, 50, 100, 200, 350, 500\}$. The SWD approximation becomes tighter with an increased number of projections, which we see translating on all three tasks. We also note that above a certain threshold, i.e. $L \approx 200$, the gains in performance from increasing $L$ are minimal.

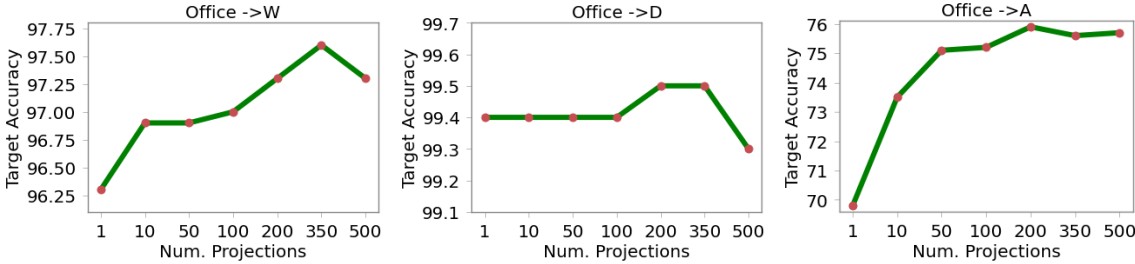

Figure 2: Performance for different numbers of latent projections used in the SWD on Office-31.

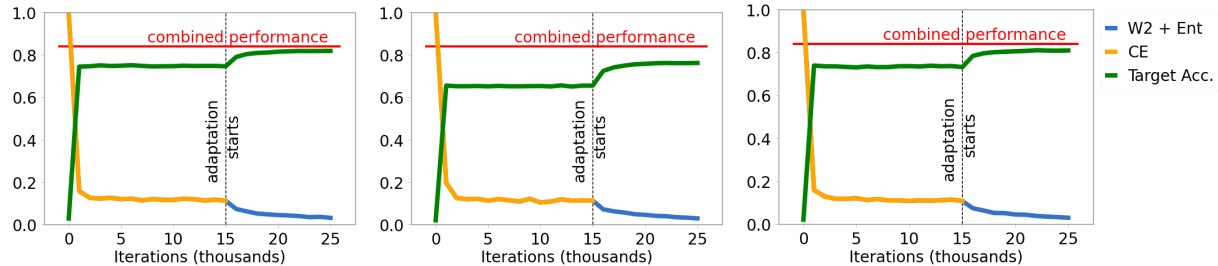

Figure 3: Effect of the adaptation process on the *Office-home* dataset: from left to right, we consider *Art, Clipart* and *Product* as the source domains, and *Real World* as the target domain.

In Figure 3 we explore the behavior of our adaptation strategy with respect to a *Office-home* task. For each of the three source domains, we observe an increase in target accuracy once adaptation starts, which is in line with our previous results. Note this increase in target accuracy also correlates with the minimization of the SWD and entropy losses. We additionally note that the combined multi-source performance using all three source domains outperforms the three SUDA performances. The biggest difference is observed for the *Clipart* trained model, which exhibits the highest discrepancy from the target domain *Real World*.

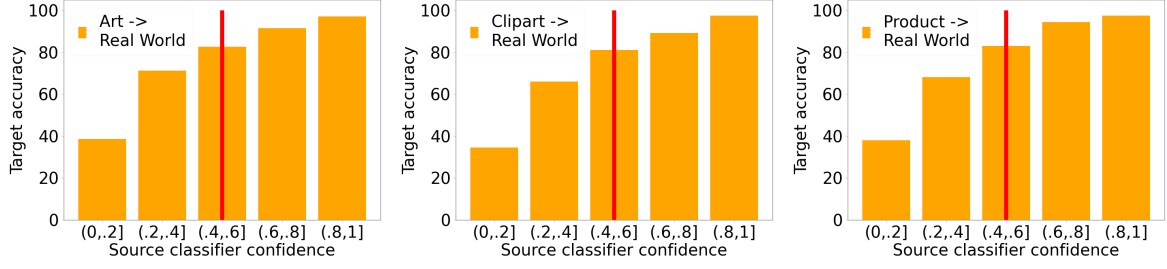

Figure 4: Prediction accuracy on *Office-home* target domain tasks under different levels of source model confidence, and our choice of $\lambda$. Target predictions above this threshold attain high accuracy.

The confidence threshold $\lambda$ controls the assignment of mixing weights $w_k$. For each source domain, the number of target samples with confidence greater than $\lambda$ is recorded, and these normalized values produce $w_k$. In order to determine whether a certain value of $\lambda$ leads to a satisfactory choice of mixing weights, it is important to determine whether the high confidence samples are indeed correctly predicted. Figure 4 provides the prediction accuracy on target domain samples on the *Office-home* dataset for different confidence ranges. We consider 5 different confidence probability ranges: $[0-20, 20-40, 40-60, 60-80, 80-100]$. We observe low-confidence predictions offer poor accuracy for the target domain. For example, in cases when the confidence is less than 0.2, prediction accuracy is below 40%. Conversely, for target samples with a predicted confidence greater than .6, we observe accuracy of more than 90% on all the three tasks of *Office-home*. This experiments supports our intuition that the amount of high confidence target samples can be used as a proxy for the domain mixing weights $w_k$. We also note the amount of high confidence samples is calculated using the source only models, as adaptation artificially increase confidence across the whole dataset.

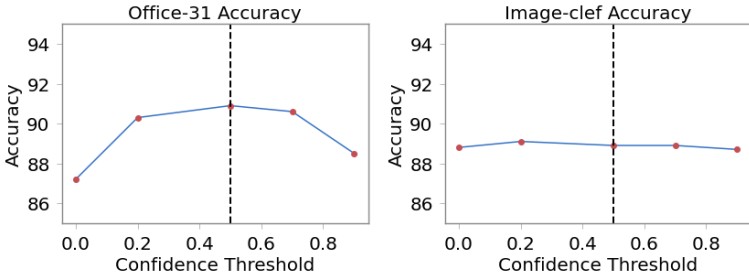

Figure 5: Results on the *Office-31* and *Image-clef* datasets for different values of the confidence parameter $\lambda$. The dotted line corresponds to $\lambda = .5$ used for reporting results in Table 1.

We further investigate performance in regards to the $\lambda$ parameter. While in Figure 5 we observe a target accuracy increase correlated to higher levels of classifier confidence, the amount of high confidence samples proportional to dataset size is equally important for an appropriate choice of confidence threshold. Setting the $\lambda$ parameter too high may lead to mixing weights that do not capture model behavior on the whole target distribution, just on a small subset of samples, leading to degraded performance. Conversely, a low value of $\lambda$ will lead to results that are equivalent to uniformly combining predictions. Figure 5 portrays both these behaviors on the *Office-31* and *Image-clef* datasets. We observe our choice of $\lambda = .5$ is able to obtain best performance on the *Office-31* dataset, and close to best performance on the *Image-clef* dataset. We also note the choice of $\lambda$ is relatively robust, as values in the interval $[.2, .7]$ offer similar performance.

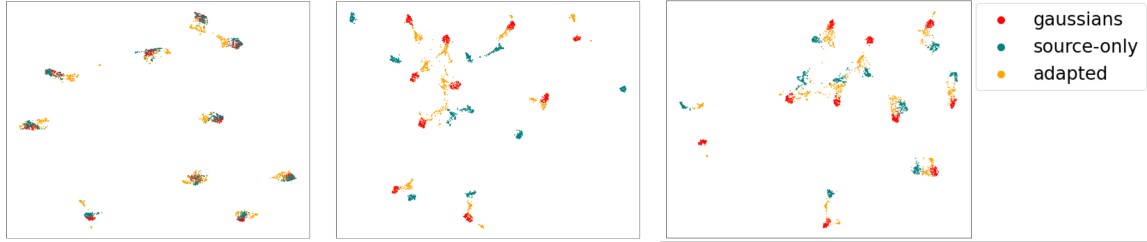

Figure 6: UMAP latent space visualization for *Office-caltech* with *Amazon* as the target. Sources in order: *Caltech, DSLR*, and *Webcam*. Adaptation shifts target embeddings towards the GMM distribution.

Our approach attempts to minimize the distributional distance between target embeddings and GMM estimations of source embeddings. We provide insight into this process in Figure 6, where we reduce the data representation dimension to two using UMAP (McInnes et al. (2018)). We display GMM samples, target latent embeddings before adaptation, and target latent embeddings post-adaptation. For each source domain, the adaptation process reduces the distance between target domain embeddings (yellow points) and the GMM samples (red points). This empirically validates the theoretical justification for our algorithm. Given classifiers trained on the source domains are able to generalize on the GMM samples as a result of pretraining, we conclude that source-specific domain alignment translates to an improved collective performance.

## 7 Conclusion

We develop a privacy-preserving MUDA algorithm based on the assumption that an input distribution is mapped into a multi-modal distribution in an embedding space. We maintain cross-domain privacy by minimizing the SWD loss between an intermediate GMM distribution and the target domain distribution in the latent embedding. We then combine the source-specific models according to their reliability. We provide theoretical analysis to justify our algorithm. Our experiments demonstrate that our algorithm performs favorably against SOTA MUDA algorithms using five UDA benchmarks while preserving privacy. Future direction includes considering setting where the target domain shares different classes with each of sources.

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

# A   Appendix

## A.1   Proof of Theorem 5.1

We offer a proof for Theorem 5.1 from the main paper. Consider the following results.

**Theorem 2.** *Theorem 2 from  Redko & Sebban (2017)*

*Let $h$ be the hypothesis learnt by our model, and $h^*$ the hypothesis that minimizes $e_{\mathcal{S}} + e_{\mathcal{T}}$. Under the assumptions described in our framework, consider the existence of $N$ source samples and $M$ target samples, with empirical source and target distributions $\hat{\mu}_{\mathcal{S}}$ and $\hat{\mu}_{\mathcal{T}}$ in $\mathbb{R}^d$. Then, for any $d' > d$ and $\zeta < \sqrt{2}$, there exists a constant number $N_0$ depending on $d'$ such that for any $\xi > 0$ and $\min(N, M) \geq N_0 \max(\xi^{-(d'+2)}, 1)$ with probability at least $1 - \xi$, the following holds:*

$$
\begin{aligned}
e_{\mathcal{T}}(h) \leq & e_{\mathcal{S}}(h) + W(\hat{\mu}_{\mathcal{T}}, \hat{\mu}_{\mathcal{S}}) + \\
& \sqrt{\left(2 \log(\frac{1}{\xi})/\zeta\right)} \left(\sqrt{\frac{1}{N}} + \sqrt{\frac{1}{M}}\right) + e_{\mathcal{C}}(h^*)
\end{aligned}
\tag{7}
$$

The above theorem provides an upper bound on the target error with respect to the source error, the distance between source and target domains, a term that is minimized based on the number of samples, and a constant $e_{\mathcal{C}} = e_{\mathcal{S}}(h^*) + e_{\mathcal{T}}(h^*)$ describing the performance of an optimal hypothesis on the present set of samples.

We adapt the result in Theorem 2 to provide an upper bound in our multi-source setting. Consider the following two results.

**Lemma 1.** *Under the definitions of Theorem 2*

$$
W(\hat{\mu}_{\mathcal{S}}, \hat{\mu}_{\mathcal{T}}) \leq W(\hat{\mu}_{\mathcal{S}}, \hat{\mu}_{\mathcal{P}}) + W(\hat{\mu}_{\mathcal{P}}, \hat{\mu}_{\mathcal{T}})
\tag{8}
$$

*where $\hat{\mu}_{\mathcal{P}}$ is the GMM distribution learnt for source domain $\mathcal{S}$.*

*Proof.* As $W$ is a distance metric, the proof is an immediate application of the triangle inequality. $\qquad\square$

**Lemma 2.** *Let $h$ be the hypothesis describing the multi-source model, and let $h_k$ be the hypothesis learnt for a source domain $k$. If $e_{\mathcal{T}}(h)$ is the error function for hypothesis $h$ on domain $\mathcal{T}$, then*

$$
e_{\mathcal{T}}(h) \leq \sum_{k=1}^{n} w_k e_{\mathcal{T}}(h_k)
\tag{9}
$$

*Proof.* Let $p(X) = \sum_{k=1}^{n} w_k f_k(X)$ with $\sum w_k = 1, w_k > 0$ be the probabilistic estimate returned by our model for some input $X$, and let $y$ be the label associated with this input. The proof for the Lemma proceeds as follows

$$e_{\mathcal{T}}(h) = \mathbb{E}_{(X,y)\sim\mathcal{T}}\mathcal{L}_{ce}(p(X), \mathbb{1}_y)$$
$$= \mathbb{E}_{(X,y)\sim\mathcal{T}} - \log p(X)[y]$$
$$= \mathbb{E}_{(X,y)\sim\mathcal{T}} - \log(\sum_{k=1}^{n} w_k f_k(X)[y])$$
$$\leq \mathbb{E}_{(X,y)\sim\mathcal{T}} \sum_{k=1}^{n} w_k(-\log f_k(X)[y]) \text{ Jensen's Ineq.}$$
$$= \sum_{k=1}^{n} w_k \mathbb{E}_{(X,y)\sim\mathcal{T}}\mathcal{L}_{ce}(f_k(x), \mathbb{1}_y)$$
$$= \sum_{k=1}^{n} w_k e_{\mathcal{T}}(h_k)$$

$\square$

We now extend Theorem 2 as follows

**Theorem 3.** *Multi-Source unsupervised error bound (Theorem 5.1 from the main paper)*

*Under the assumptions of our framework and using the definitions from Theorem 2*

$$e_{\mathcal{T}}(h) \leq \sum_{k=1}^{n} w_k(e_{\mathcal{S}_k}(h_k) + W(\hat{\mu}_{\mathcal{T}}, \hat{\mu}_{\mathcal{P}_k}) + W(\hat{\mu}_{\mathcal{P}_k}, \hat{\mu}_{\mathcal{S}_k}) + \tag{10}$$
$$\sqrt{(2\log(\frac{1}{\xi})/\zeta)}(\sqrt{\frac{1}{N_k}} + \sqrt{\frac{1}{M}}) + e_{\mathcal{C}_k}(h_k^*))$$

*where $\mathcal{P}_k$ is the sample GMM distribution learnt for source domain $k$, $N_K$ is the sample size of domain $k$, $\mathcal{C}_k$ is the combined error loss with respect to domain $k$, and $h_k^*$ is the optimal model with respect to this loss.*

*Proof.*

$$e_{\mathcal{T}}(h) \leq \sum_{k=1}^{n} w_k e_{\mathcal{T}}(h_k) \text{ From Lemma 2}$$
$$\leq \sum_{k=1}^{n} w_k(e_{\mathcal{S}_k}(h_k) + W(\hat{\mu}_{\mathcal{T}}, \hat{\mu}_{\mathcal{S}_k}) +$$
$$\sqrt{(2\log(\frac{1}{\xi})/\zeta)}(\sqrt{\frac{1}{N_k}} + \sqrt{\frac{1}{M}}) + e_{\mathcal{C}_k}(h_k^*)) \text{ by Theorem 2}$$
$$\leq \sum_{k=1}^{n} w_k(e_{\mathcal{S}_k}(h_k) + W(\hat{\mu}_{\mathcal{T}}, \hat{\mu}_{\mathcal{P}_k}) + W(\hat{\mu}_{\mathcal{P}_k}, \hat{\mu}_{\mathcal{S}_k}) +$$
$$\sqrt{(2\log(\frac{1}{\xi})/\zeta)}(\sqrt{\frac{1}{N_k}} + \sqrt{\frac{1}{M}}) + e_{\mathcal{C}_k}(h_k^*)) \text{ by Lemma 1}$$

$\square$

## A.2 Experimental parameters

We use the Adam optimizer with source learning rate of $1e-5$ for each source domain for all datasets. Target learning rates are chosen between $1e-5$ and $1e-7$ for adaptation. The number of training iterations

and adaptation iterations differs per dataset: Office-31 (12k, 48k), Domain-net (80k, 160k), Image-clef (4k, 3k), Office-home (40k, 10k), Office-CalTech (4k, 6k). The training batch size is either 16 or 32, with little difference observed between the two. The adaptation batch size is usually chosen around $10\times$ the number of classes for each dataset, to ensure a good class representation when minimizing the SWD distance. The network size is the same across all datasets, with the SWD minimization space being 256 dimensional. The above mentioned parameters are also provided in the *config.py* file in the codebase.

### A.3  Additional Results

We extend the runtime results from *Section 6.3* of the main paper to the *Domain-Net* dataset. As seen in Figure 3, the result in Figure 7 share a similar trend. After the start of the adaptation process target accuracy improves for each source trained model. Additionally, pooling information from each of the five source domains leads to improved overall predictive quality of the model.

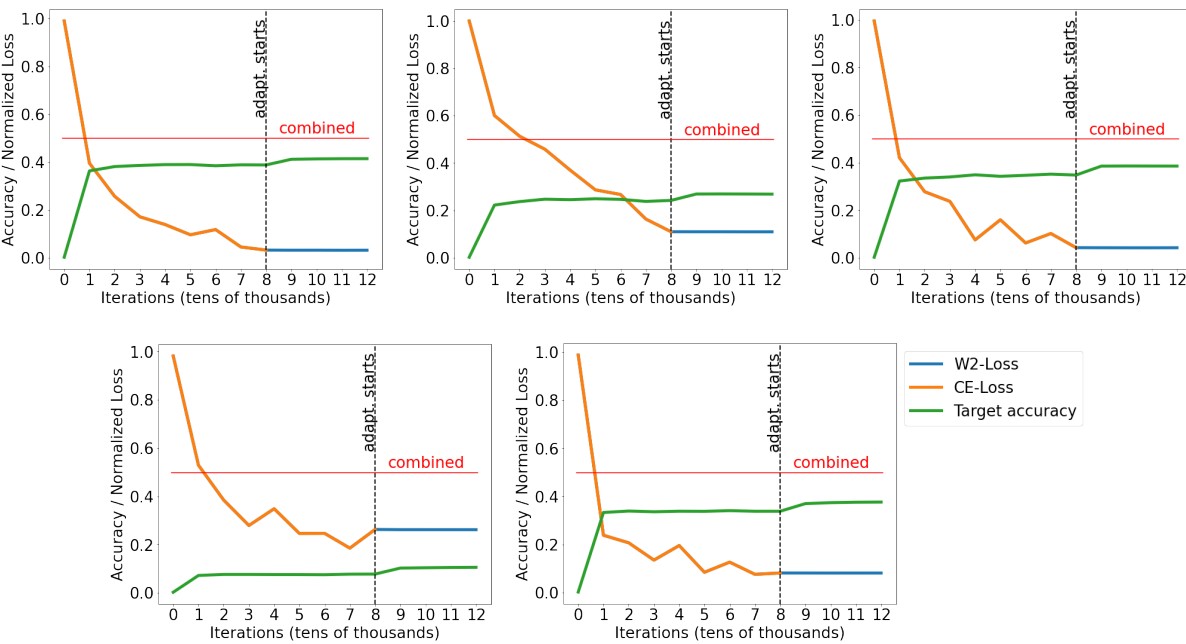

Figure 7: Effect of the adaptation process on the *Domain-Net* dataset, where sketch is the target. Sources are in order *Clipart, Infograph, Painting, Quickdraw, Real.*

In Table 7 we report results for different choices of the $\gamma$ parameter on the Office-31 dataset. The main results in Table 1 are generated for the choice of $\gamma = .02$ . We test a wider range of $\gamma$ values to identify how robust this parameter needs to be. We observe large values of $\gamma$ put too much emphasis on the entropy loss, harming the optimal transport based distribution matching. Small values of $\gamma$ share a similar problem, as we lose the soft-clustering benefit provided by the entropy loss. Overall we notice different tasks have different high performing $\gamma$ ranges . For example, on the $\rightarrow D$ task, any choice of the parameter greater than 0.02 leads to best performance, while on the $\rightarrow A$ task $\gamma$ less than 0.04 offer a competitive range of values. While setting task specific $\gamma$ values may lead to improved performance, we show that the more robust setting where we choose the same $\gamma$ per dataset still works reasonably well in practice.

We give justification for the beneficial effect of using all the available source domains for inference. In Table 8 we present results obtained from successively adding source domains to our ensemble, for the Office-home dataset. We also include single best performance as a baseline, representing the highest target accuracy obtained across source domains. The rows of the table correspond to the four MUDA problems considered for Office-home, while the columns correspond to the number of source domains considered in ensembling. For example, the $ACP \rightarrow R$ task with 2 sources considers the problem $AC \rightarrow R$. Similar to Figures 3 and

| Method | →D | →W | →A | Avg. |
|---|---|---|---|---|
| $\gamma$=1 | **99.8** | 97.6 | 64.8 | 87.4 |
| $\gamma$=.04 | **99.8** | **98.7** | 72.8 | 90.4 |
| $\gamma$=.03 | **99.8** | 98.4 | 74.2 | 90.8 |
| $\gamma$=.02 | **99.8** | 98.5 | 75.4 | **91.2** |
| $\gamma$=.01 | 98.8 | 97.8 | **76.3** | 91 |
| $\gamma$=1e-3 | 89.3 | 92.4 | 74.4 | 85.4 |
| $\gamma$=1e-4 | 87.4 | 90.7 | 74.2 | 84.1 |

Table 7: Performance analysis for different values of $\gamma$

7, we observe that ensembling is superior to single best performance on all tasks. Additionally, our mixing strategy proves to be robust with respect to negative transfer, as adding new domains is always beneficial to the reported performance.

| Method | Single best | First src. domain | First 2 src. domains | All 3 src. domains |
|---|---|---|---|---|
| $ACP \rightarrow R$ | 81.1 | 81.2 | 82.3 | **83.7** |
| $ACR \rightarrow P$ | 83.1 | 77.1 | 77.9 | **83.3** |
| $APR \rightarrow C$ | 59.2 | 58.3 | 60.1 | **60.9** |
| $PCR \rightarrow A$ | 67.2 | 67.3 | 67.6 | **68.2** |

Table 8: Performance analysis when source domains are introduced sequentially.

We investigate the representation quality of the GMM distribution as a surrogate for the source distribution. Note that having GMMs that are good approximation of the source latent features is crucial for our approach. In Figure 8, we present visualized data representations for the estimated GMMs and the source domain distributions for the *Image-clef* dataset. We note that for both source domains, their latent space distributions after pretraining are multi-modal distributions with 12 modes, each corresponding to one class. This observation confirms that we can approximate the source domain distribution with a GMM with 12 modes. We also note that for both source domains the estimated GMM distribution offers a close approximation of the original source distribution. This experiment empirically validates that the third term in Eq. 4 is small in practice and we can use these as intermediate cross-domain distributions.

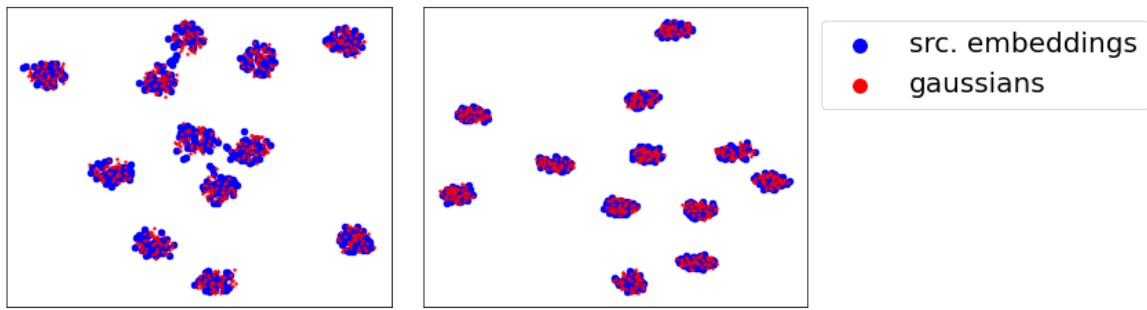

Figure 8: Source and GMM embeddings for the *Image-clef* dataset with *Pascal* and *Caltech* as sources. For both datasets, the GMM samples closely approximate the source embeddings.

We additionally analyze the latent source approximation used for adaptation, and compare it to the scenario when fine tuning the model on the source domain can be done after adaptation starts. We present these findings in Figure 9. When privacy is not a concern and we have access to all source data, there is no need to approximate the source distribution during adaptation, as we can just directly use the source samples. For empirical exploration, results on how the latent distribution changes when source access is permitted during adaptation (corresponding to the SS + SW case in Table 3) is presented in Figure 9. We notice that as training progresses, the means slightly shift. Visually however, the change compared to using the latent

space only after source training is negligible. This supports the idea that the GMMs learned at the end of source training will provide a sufficient approximation of the source distribution.

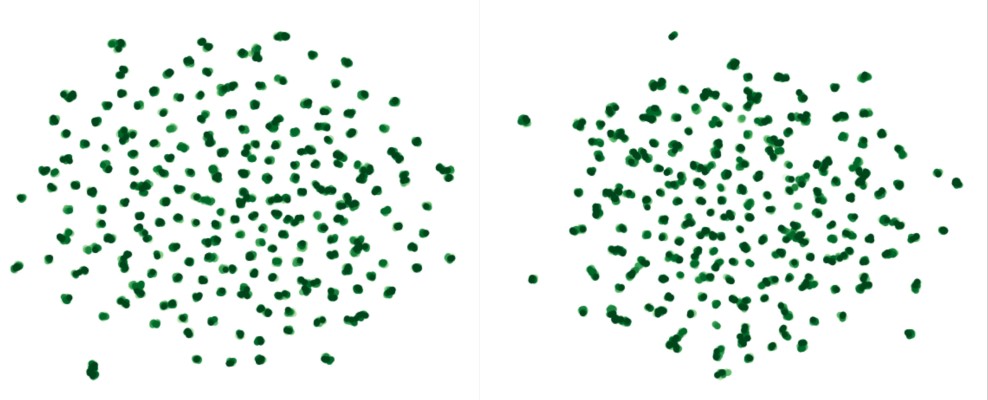

Figure 9: Latent distributions for two domains of the Office-31 dataset when considering the $D \rightarrow A$ and $W \rightarrow A$ tasks. Each color gradient represents a latent feature distribution snapshot of the source domains. Darker colors correspond to later training iterations, lighter colors to earlier iterations.

