# OpenReview forum: "Secure Domain Adaptation with Multiple Sources"
_TMLR — Accepted by TMLR_

### Review · Reviewer_TfHu · 2022-07-03

**Summary Of Contributions:**

In this paper, the authors propose a new multi-source unsupervised domain adaptation (MUDA) method that does not require source data when training the model in the target domain, thus avoiding issues such as data privacy. The proposed method performs domain adaptation in each source and target domain. The final output is obtained by computing a weighted sum of the resulting adapted models for each source domain based on their confidence levels. During the adaptation of each source and target domain, the distribution of the source domain data is modeled as a mixture of Gaussians to avoid direct access to the source domain data, and domain adaptation is performed to align the estimated source distribution with the target distribution. The authors also show that the proposed method minimizes the upper bound on errors in the target domain through theoretical analysis. Furthermore, extensive experiments verify the effectiveness of the proposed method.


**Broader Impact Concerns:**

There are no concerns on the ethical implications of the work.

**Requested Changes:**

1. Many single-source SFUDA methods have been proposed. This paper needs a comparison with the ensemble models of existing single-source SFUDA methods that adapt the target model to each source domain.
2. This paper should show how much performance improvement the proposed SFUDA (a part of the proposed model w/o ensemble) between each source and target has over other single-source SFUDA methods.
3. Since other multi-source SFUDA methods exist besides DECISION [a], this paper needs at least a qualitative comparison with them (e.g., [b]).
4. Methods for domain adaptation by estimating the probability density distribution of the source domain and aligning that distribution with the target distribution have already been proposed [c, d, e, f, g]. This paper should compare the proposed one at least qualitatively with them.
5. In training the target model, the initial parameter values are copied from the parameters of the source model. The reviewer would like to know if there are any privacy issues.
6. This paper should describe the values of the hyperparameters when the results of TABLE 1 are obtained. The readers would like to know how much optimization is done for each dataset.


- [a] Ahmed, Sk Miraj, et al. "Unsupervised multi-source domain adaptation without access to source data." Proceedings of the IEEE/CVF Conference on Computer Vision and Pattern Recognition. 2021.
- [b] Dong, Jiahua, et al. "Confident anchor-induced multi-source free domain adaptation." Advances in Neural Information Processing Systems 34 (2021): 2848-2860.
- [c] (align with GAN-generated source-like data) Kurmi, Vinod K., Venkatesh K. Subramanian, and Vinay P. Namboodiri. "Domain impression: A source data free domain adaptation method." Proceedings of the IEEE/CVF Winter Conference on Applications of Computer Vision. 2021.
- [d] (GMM-based) Yeh, Hao-Wei, et al. "Sofa: Source-data-free feature alignment for unsupervised domain adaptation." Proceedings of the IEEE/CVF Winter Conference on Applications of Computer Vision. 2021.
- [e] (GMM-based) Yang, Baoyao, et al. "Model-Induced Generalization Error Bound for Information-Theoretic Representation Learning in Source-Data-Free Unsupervised Domain Adaptation." IEEE Transactions on Image Processing 31 (2021): 419-432.
- [f] (GMM-based, means and variances are computed by weights of source classifier) Tian, Jiayi, et al. "Vdm-da: Virtual domain modeling for source data-free domain adaptation." IEEE Transactions on Circuits and Systems for Video Technology (2021).
- [g] (GMM-based, means and variances are computed by target features from source model) Ding, Ning, et al. "Source-Free Domain Adaptation via Distribution Estimation." Proceedings of the IEEE/CVF Conference on Computer Vision and Pattern Recognition. 2022.


**Strengths And Weaknesses:**

Strength

1. This paper addresses the multi-source UDA problem, where source data is not required when training models in the target domain. This is a practical approach for privacy-sensitive issues such as medical data.
2. The model adaptation between the source and target is performed independently for each source domain so that if the source domain increases or decreases, only the changed source domain needs to be adapted.
3. Theoretical analysis has been performed and shows that the proposed algorithm minimizes the upper bound of errors in the target domain.
4. The effectiveness of the proposed method is verified using five standard datasets in UDA.
5. The proposed algorithm is simple and easy to implement.
6. The paper is well written and easy to understand.

Weakness
1. The proposed method adapts single-source Source-free UDA (SFUDA) to each source domain and computes an ensemble of them. Therefore, the novelty of the proposed method is not significant in the general framework.
2. Source-free UDA by estimating the probability density distribution of the source domain and aligning that distribution with the target distribution has already been proposed  [c, d, e, f, g in the next section], so the method is not novel at this point.
3. The target model copies the parameters of the source model in the early stages of learning, which is somewhat problematic from a privacy perspective since the source's post-training parameters are accessible.
4. The number of models increases with the number of source domains, making model size inefficient. In addition, when data from the target domain is input, the inference must be performed in proportion to the number of source models, which increases inference time in proportion to the # of source domains.
5. In the experimental results in Table 1, the method DECISION performs better than the proposed method, especially for more complex data sets such as Office-home and DomainNet.

---

> ### Author Response · Authors · 2022-08-22
> **Response to the reviewer**
>
> Dear reviewer,
>
> Thank you for your time and feedback. We have tried to revise the manuscript according to the collective reviews we received. Since our responses include figures and tables, presenting them in a comment-fashion is not possible. For this reason, we used latex and included our responses as a supplementary material. Please download the supplementary material and find our responses. We hope we can use the remaining time to improve the manuscript with you new feedback.
>
> Thank you,
> Our team

---

### Review · Reviewer_Q4QX · 2022-07-04

**Summary Of Contributions:**

The paper proposes an algorithm for multi-source unsupervised domain adaptation (MUDA) for the scenarios when source and target domain data is not available at the same time due to privacy and security concerns.

The algorithm starts by learning a model for each source domain separately and assuming the representation space of the encoder to be a gaussian mixture model estimates their means and variances, separately for each source domain (regardless of other sources and the target).

To address the concerns of privacy the domain adaptation part is done based on the learned means and variances of the individual source domain and the target domain.

After the adaptation part of each source and the given target is complete the final classifier is obtained by using a weighted combination of the individual classifier.

A confidence-based technique is used to combine the classifiers for prediction of a point at test time. Using ablation studies the paper shows confidence-based scheme to be the best among different possible combination schemes.

Extensive experiments and ablation tests have been performed in the paper to clarify the effectiveness of the proposed algorithm and the different design choices used.
The paper also shows that the proposed algorithm is theoretically motivated and minimizes the upper bound on the target domain error.



**Broader Impact Concerns:**

-

**Requested Changes:**

Answers to questions raised in the Weakness/Concern section.

It would be great to see some empirical results or discussion on the differences between the performances of the algorithms with and without privacy to estimate how much performance is lost by enforcing privacy constraints. I would also be interested in seeing a discussion on the difference in the upper bound on the target domain error with and without privacy constraints.

Additional discussion on why the GMM choice is suitable and practical would also be beneficial to the audience trying to incorporate privacy in their methodology.

Lastly, some discussion about how privacy leaks are measured should be added. It would be beneficial to add previous works on this topic too if there are any that the authors are aware of.

Minor:
1. Change Eq. 3 to add L and refer to it on line 14 of Alg. 1
2. Typo in spelling above eq 5.


**Strengths And Weaknesses:**

Overall the paper addresses an important problem in MUDA and proposes an effective algorithm to solve the problem.
The paper extensively ablates different hyperparameters used in the algorithm and presents a nice comparison that justifies the different design choices.
The online nature of the algorithm, in that it can incorporate new source domains without having to retrain the entire pipeline gives a computational advantage to the method.

However, I would like to get an idea of how worse the algorithm performs in comparison to the scenario when privacy is not enforced. In particular, how different is the accuracy of the model on the target domain both theoretically and empirically, if you could use all sources and the target simultaneously?

Another concern is around the assumption of feature space learned using individual sources to be a gaussian mixture model. Although the last paragraph above the conclusion, provides an explanation of why this works, I am more interested in understanding how practical is this assumption. Does this hold for most datasets or is it being used here to make the computation easy.

Related to the previous concerns, the algorithm does not share data across the domains but only the estimated means and variances.
How different are the means and variances of the domains estimated with and without privacy concerns?
Does the privacy-preserving MUDA suffer significantly in comparison to vanilla MUDA algorithms? If so, is it possible to narrow this gap if we computed the means and variances of all source domains (just as is done in lines 3 and 4 of Alg 1) but used them together to do the adaptation step? Particularly, instead of getting \theta_k^A on line 6 for every k, we only got one \theta_A = Adapt(all \theta_k, all A_k, T, L, \gamma).

Lastly, is there a direct way of measuring privacy leaks? Is it possible that sharing means and variances also lead to leaks in privacy?

---

> ### Author Response · Authors · 2022-08-22
> **Response to the reviewer**
>
> Dear reviewer,
>
> Thank you for your time and feedback. We have tried to revise the manuscript according to the collective reviews we received. Since our responses include figures and tables, presenting them in a comment-fashion is not possible. For this reason, we used latex and included our responses as a supplementary material. Please download the supplementary material and find our responses. We hope we can use the remaining time to improve the manuscript with you new feedback.
>
> Thank you,
> Our team

---

> > ### Comment · Reviewer_Q4QX · 2022-09-15
> > **Response to authors**
> >
> > I thank the authors for their detailed responses to my questions. While some of my concerns have been addressed, I still have questions about the difference between SMUDA and MUDA without privacy constraints. In the response, authors have shown that SMUDA leads to the same performance compared to different algorithms for MUDA. This seems counterintuitive to me since I would have expected that without imposing privacy constraints accuracy of MUDA should be much better. There could be two possible reasons for this behavior, firstly, MUDA methods without any privacy constraints cannot use the information available to them effectively, or second, there is some privacy leak in SMUDA which makes it achieve very similar performance to the case with no privacy constraints (perhaps I have misunderstood something in the experiment and results in Table 3). Lastly, do the results in Table 3 for Office-31 hold for other datasets too? It would be great if the authors can clarify this part in some detail.

---

> > > ### Author Response · Authors · 2022-09-17
> > > **Response to concerns about Table 3**
> > >
> > > Dear Reviewer
> > >
> > > Thank you for your continual engagement. We are glad that we have been able to address some of your concern. Please see our response to address your remaining concern about results in Table 3.
> > >
> > > 1. Please note adding privacy and security constraints can lead to degraded performance. If you check Table 3, you can see that our method is not outperforming the SS+SW baseline that is a version of our method that uses the source data for computing both the SWD and the supervised losses. We can also think of scenarios, where using our method can  increase the performance, despite the privacy constraint. For example, if we have some outlier samples in the source data, our approach allows to discard outlier data samples because the estimated Gaussian distribution is computed by discarding discarding outliers, i.e., the mean and the covariance matrix of the Gaussian are less sensitive with respect to a few number of outliers. Hence, our method can potentially improve the performance. We agree this explanations may look counterintuitive, but please note that our method benefits from the source domain data and the shared model encodes the information gained from the source domain. When we use this model along with the intermediate estimation of distribution, we only lose a small amount of information that can be learned from the source domain and this feature is reason behind the success of our method.
> > >
> > > 2. We generated Table 3 for the Office-Home dataset as follows to expand our empirical exploration:
> > >
> > >
> > > Method:  ->A,  ->C,   ->P,  ->R,    Avg.
> > >
> > > SW:       68.9   60.8  83.3  83.4   74.1
> > >
> > > SC:       69.6   62.9  85.3  84.7   75.6
> > >
> > > SW+SS:    68.5   61.0  83.8  83.8   74.3
> > >
> > > SC+SW+SS: 68.8   62.7  85.1  84.5   75.3
> > >
> > > SMUDA:    69.2   61.1  83.2  83.5   74.3
> > >
> > >
> > > Note that Office-Home has more classes (65 compared to 31) and larger datasets compared to Office-31. We observe similar results to those of Office-31 case (Table 3 in the revision). The primary difference  is that the source combined performance is able to outperform the other methods, whereas on Office-31 it was slightly trailing. The relative performance of the other variations remains the same, i.e. SW $\sim$ SW+SS $\sim$ SMUDA, SC $\sim$ SC+SW+SS.  We will include these results along with Table 3 in the next revision.
> > >
> > > Finally, we would like to highlight that coming up with mathematical proofs is always challenging when large-scale neural networks are used for modeling.
> > >
> > > Thank you for your time and consideration,
> > >
> > > Our team

---

> > > > ### Comment · Reviewer_Q4QX · 2022-09-26
> > > > **Response to authors**
> > > >
> > > > I thank the authors for their prompt response and for presenting results on another dataset with regard to my concern about the difference in the performance of the method with and without privacy constraints. The results of the two datasets show that SMUDA performs quite similarly (only about a 2% decrease in accuracy) to MUDA without privacy constraints. I find this result quite surprising and analyzing how the target domain performance decreases when privacy is enforced could further strengthen the contributions of the paper. However, since this paper is demonstrating these findings empirically I think strengthening the empirical evidence by evaluating more datasets as well as including a discussion on why accuracy does not degrade much with SMUDA could be useful for the readers.

---

> > > > > ### Author Response · Authors · 2022-10-01
> > > > > **Results for additional datasets**
> > > > >
> > > > > Dear Reviewer,
> > > > >
> > > > > We thank you for your continual engagement. Following your advice, we performed experiments on two more datasets as follows:
> > > > >
> > > > >
> > > > > Results for the Office-caltech Dataset
> > > > >
> > > > > Method:          ->W,   ->C,   ->D,  ->A,  Avg.
> > > > >
> > > > > SW:                  99.4  96.9  93.9  95.9  96.5
> > > > >
> > > > > SC:                   99.7  96.8  94.1  96.0  96.6
> > > > >
> > > > > SW+SS:           99.7  97.4  94.1  96.0  96.8
> > > > >
> > > > > SC+SW+SS:    99.6  97.2  93.3  95.9  96.5
> > > > >
> > > > > SMUDA:           99.3  97.6  93.9  95.9  96.6
> > > > >
> > > > >
> > > > > Results for the mage-clef Dataset
> > > > >
> > > > > Method:          ->P,   ->C,   ->I,  Avg.
> > > > >
> > > > > SW                 79.5  95.2  91.3  88.6
> > > > > SC                  79.8  96.6  94.2  90.2
> > > > > SW+SS          79.4  95.6  91.8  88.9
> > > > > SC+SS+SW   79.5  95.2  91.3  88.6
> > > > > SMUDA          79.4  96.9  93.9  90.1
> > > > >
> > > > >
> > > > > As it can be seen, these additional results are not significantly different from previous results. We agree with you that these results might look surprising, but we think they may not be as unexpected as they look. The reason is that we are not giving up source data. When the model is trained on the source domain, information that can be learned from the source domain data is encoded in the network weights. Following your advice, we will add these new results in the manuscript and add a paragraph. We hope that we have been able to address your concerns.
> > > > >
> > > > > Thank you,
> > > > >
> > > > > Our team

---

### Review · Reviewer_7zKM · 2022-08-19

**Summary Of Contributions:**

This paper studies the task of multi-source unsupervised domain adaptation (MUDA) where the data of each source domain are assumed to be unavailable to keep privacy during adaptation. For this purpose, the authors propose to learn the latent representation of each source domain via GMM, and then apply the surrogate distribution for the alignment between the source and target domain. The adaptation for each source domain is carried out independently and the predictions of target samples are obtained via a hand-crafted convex combination of logits of all adapted models. Experiments are conducted on several benchmarks with certain ablations studies performed as well.

**Broader Impact Concerns:**

None.

**Requested Changes:**

See the weakness part above.

**Strengths And Weaknesses:**

Strengths:

1.This paper is over-all well written and the references seem sufficient.

2.It is sort of variable to enable the independent adaptation from each source domain.

3.Necessary experiments are conducted to support the proposed idea.

Weaknesses:

1.The biggest concern is that the novelty is weak, compared to a related work DECISION （Ahmed et al. 2021), which also explores MUDA without the access of source domain data. The only difference from DECISION is that this work claims the benefit of independent adaptation from each source domain. This results in the next issue.

2.The motivation of performing independent adaptation is still unclear or not well-grounded. The authors claim that this setting is constrained yet practical, which seems quite subjective. DECISION assumes the access of all pretrained models where the privacy of all source data is still maintained. More importantly, DECISION allows the end2end finetuning on the target data (including the retraining of the feature extractors of all source domains and the combination weights), leading to better performance than the proposed model in this paper (as illustrated in Table c, d, e). While the authors state that this work allows for accumulative learning from several domains when new source domain comes, there is no experimental justification on this point.

3.GMM is still a weak surrogate distribution for more practical and complicated datasets. It is expected that GMM is unable to encode the sufficient patterns within the input data. What if we change GMM to other choices? Such as GAN?

4.As stated above, the combination weights (Eq. 6) are hand-crafted, leading to suboptimal performance compared to DECISION where the feature extractor and the weights are learned simultaneously and end to end.

Writing issues:

1.	The abstract and introduction require rewriting, as main important features such as GMM and the difference compared to DECISION are not included.

2.	Eq. (4) is unclear. More details needed.

3.	Eq. (6) is confusing. The function f should output a label other than the logit. And what about multi-class classification? Not just binary classification.

---

> ### Author Response · Authors · 2022-08-22
> **Response to the reviewer**
>
> Dear reviewer,
>
> Thank you for your time and feedback. We have tried to revise the manuscript according to the collective reviews we received. Since our responses include figures and tables, presenting them in a comment-fashion is not possible. For this reason, we used latex and included our responses as a supplementary material. Please download the supplementary material and find our responses. We hope we can use the remaining time to improve the manuscript with you new feedback.
>
> Thank you,
> Our team

---

### Review · Reviewer_rxyr · 2022-08-22

**Summary Of Contributions:**

This paper proposes a multi-source unsupervised domain adaptation (MUDA) method targeting the use case, where (both source and target) datasets are distributed among independent entities and can not be shared across some central accessibility due to privacy protection. To adapt to each source domain while preserving privacy, the proposed method chooses to use a model ensemble tactic, which approximates the distributions of the source domains in the embedding space and uses these distributional estimations for domain alignments through Gaussian Mixture Models (GMM). More specifically, for each source domain, a model is independently trained along with the calculation of the pair of mean and covariance parameters in the latent space. The models are then adapted to pseudo-domains through the GMM distribution in the embedding space and conditional cross-entropy. The final output of the input from the target domain will be the weighted sum of the models from all the source domains based on the confidence scores of the prediction. Both theoretical and empirical results show the effectiveness of the proposed method. Regarding privacy, this method achieves using only the latent means and covariance of the estimated GMMs as the cross-domain information and data samples are never shared between any two domains during either the training or adaptation phase.

**Broader Impact Concerns:**

No concerns on the broader impact from the reviewer.

**Requested Changes:**

In general, the requested changes all correspond to the weaknesses listed above.

1. (Weakness 1 & 2) Please refer to weaknesses 1 and 2 to improve the readability of the figure or the introduction of the paper.
2. (Weakness 3) It is recommended to add some discussion and analysis on the performance of different methods against possible privacy attacks (and experiments if applicable).
3. (Weakness 5) It is recommended to add some ablation studies to study the sensitivity of the proposed method to the choice of hyper-parameter $\gamma$. If possible, please also add some discussion of the sensitivity of the proposed method to hyper-parameter choices, as was mentioned in the literature review in the first place.
4. (Weakness 6) Please add experiments to support your argument on source domain change.
5. (Weakness 7) Please add some discussions on invariant risk minimization.


**Strengths And Weaknesses:**

Strengths:

1. (Motivation) The problem addressed by the paper is indeed important in the field of MUDA and the privacy-protecting setting of the study is of practical use. The authors have also made this point clear and convincing enough through the examples of mobile keyboard predictions and medical image processing applications.
2. (Privacy) The major contribution lies in how the proposed method avoids using inter-domain statistics, such as the data, distribution, or representations in the embedding space, as many of the existing MUDA methods would do. The proposed method achieved model adaptation using only the means/covariances as the only cross-domain information.
3. (Scalability) The proposed method is also robust to changes in the accessibility of the source domains, such as increasing or decreasing the available source domains. End-to-end retraining is avoided as the model-ensemble structure made each source domain work in a plug-in mode.
4. (Theoretical) Theoretical analysis is provided to show the upper bound of the target domain error.
5. (Empirical) The experiment results show the effectiveness of the proposed method in various datasets.

Weaknesses:

1. (Presentation) There is too little information about the key technologies used in this paper disclosed in the abstract. Some high-level summarization should be mentioned to improve the readability, such as the usage of GMM, model ensembling, etc. Also, Figure 1 looks a bit scrambled with too many elements of different styles, and thus needs revision.
2. (Presentation) All the texts in the figures (legends, labels, axis ticks, annotations) are suggested to be of similar font size to that of the texts in the paper. The current font size is not consistent (see legends in Figures 4, 6, 7, etc). Also, the figures are suggested to use vector diagrams, which can avoid low-resolution problems (Figure 7).
3. (Privacy) The choice of the model ensemble is a double-edged sword. It is used to protect privacy initially. However, I am afraid in such a scenario, since the model for each source domain is separate, wouldn’t the privacy attack against the model for a particular source domain be easier to perform compared to those MUDA methods using only one model for all the source domain?
4. (Time and Memory Consumption) It is self-explanatory that the inference time, as well as the memory consumption of the proposed method, is proportionate to the number of the source domains. Take the membership inference attack (MIA) [1, 2] as an example, which is a very commonly used privacy attack. Many MIA is based on the confidence or the entropy of the output, which are both used in the proposed method. Thus, I wonder if the proposed method would suffer from MIA more compared to the non-model-ensemble-based methods.
5. (Hyper-parameter Sensitivity) The paper criticized the adversarial-learning-based UDA methods for their high dependence on optimal hyper-parameters. In this paper, it seems the newly introduced hyper-parameters $\lambda$ exert great influence on the performance as well (Figure 5) and we do not know the influence of the $\gamma$ for now.
6. It is claimed that the proposed method is friendly to source domain change, while there is no empirical evidence in the paper. I would like to see how the performance will be if more source domains are added to the well-trained model of SMUDA.
7. (Related Works) It seems the proposed method has some similarities to invariant risk minimization (IRM) [3-5], and I suggest the authors look at them as well as add some discussion on the similarity/difference.

> [1] Carlini N, Chien S, Nasr M, et al. Membership inference attacks from first principles[C]. 2022 IEEE Symposium on Security and Privacy (SP). IEEE, 2022: 1897-1914.
>
> [2] Shokri R, Stronati M, Song C, et al. Membership inference attacks against machine learning models[C]. 2017 IEEE symposium on security and privacy (SP). IEEE, 2017: 3-18.
>
> [3] Arjovsky M, Bottou L, Gulrajani I, et al. Invariant risk minimization[J]. arXiv preprint arXiv:1907.02893, 2019.
>
> [4] Ahuja K, Shanmugam K, Varshney K, et al. Invariant risk minimization games[C]. International Conference on Machine Learning. PMLR, 2020: 145-155.
>
> [5] Krueger D, Caballero E, Jacobsen J H, et al. Out-of-distribution generalization via risk extrapolation (rex)[C]. International Conference on Machine Learning. PMLR, 2021: 5815-5826.

---

> ### Author Response · Authors · 2022-08-22
> **Response Prepration**
>
> Dear Reviewer,
>
> We had received the following message in an email:
> --------------------------------------------------------------------------------
> Now that 3 reviews have been submitted for your submission Secure Domain Adaptation with Multiple Sources, all reviews have been made public. If you haven’t already, please read the reviews and start engaging with the reviewers to attempt to address any concern they may have about your submission.
>
> You will have 2 weeks to respond to the reviewers. To maximise the period of interaction and discussion, please respond as soon as possible. The reviewers will be using this time period to hear from you and gather all the information they need. In about 2 weeks (Sep 02), and no later than 4 weeks (Sep 16), reviewers will submit their formal decision recommendation to the Action Editor in charge of your submission.
> ---------------------------------------------------------------------------------
> Hence, we were not expecting a fourth review and were under the impression that we should respond to three reviews. We could surely wait for your review before preparing our response. We will try to prepare responses to your review in a timely manner.
>
> Best,
> Our team

---

> > ### Comment · Action_Editors · 2022-08-22
> > **You will have complete 2 weeks to address all reviewers' comments**
> >
> > Dear Authors,
> >
> > You will be granted a complete 2-week period to prepare for your revision and response to all four reviewers (aka the deadline is two weeks from now). Looking forward to your updates.
> >
> > Your Action Chair

---

> > > ### Author Response · Authors · 2022-08-27
> > > **Respone to the reviewer**
> > >
> > > Dear reviewer,
> > >
> > > Thank you for your time and feedback. We have tried to revise the manuscript according to the collective reviews we received. Since our responses include figures and tables, presenting them in a comment-fashion is not possible. For this reason, we used latex and included our responses as a supplementary material. Please download the supplementary material and find our responses. We hope we can use the remaining time to improve the manuscript with your new feedback.
> > >
> > > Thank you, Our team

---

> > > > ### Comment · Reviewer_rxyr · 2022-08-30
> > > > **Response to Authors**
> > > >
> > > > I thank the authors for their prompt response and all the new discussions/experiments conducted.
> > > >
> > > > In general, I think the authors have adequately addressed my concerns. The additional experiments look good and some of them are beyond my expectation. I think the authors also did a good job in discussing the relationship between UDA and IRM. It would be better if the authors can further add more discussions on the model ensemble in the revision.
> > > >
> > > > If there is no additional post from me, I think the response looks good to me.

---

> > > > > ### Author Response · Authors · 2022-08-31
> > > > > **Thank you note**
> > > > >
> > > > > Dear Reviewer,
> > > > >
> > > > > Thank you for reading the revised version and our response in a timely manner. We are glad that you found our response convincing. We will add more discussions on the model ensemble in the next revised version.
> > > > >
> > > > > Thank you,
> > > > >
> > > > > Our team

---

### Decision · Action_Editors · 2022-09-29

**Recommendation:** Accept with minor revision

**Comment:**

While all reviewers see some merits in this work, the authors' response and revision did not fully address the reviewers' comments. Therefore, I demand necessary changes and clarification on the following two action items before accepting this submission.

1. Justification on why the accuracy would not be degraded when the privacy constraint is enforced. There are known trade-offs in utility and privacy for many machine learning tasks and models, so the reported results need proper justification on why the proposed method is an exception (Please see Reviewer Q4QX's comments for details)

2. Clarification of differences to existing works (Please see Reviewer TfHu's comments for details):

The paper states, "We address the challenge of data privacy for MUDA by maintaining full privacy between pairs of source domains, and between source and target domains." as one of the main contributions. However, it is difficult to say maintaining 'full' privacy because the parameters of the source model can be accessed during model initialization. SFUDA methods that do not access source model parameters have already been proposed (e.g., [d, e]). In this respect, the paper may be overstating its contribution.

The second contribution is that "We propose an efficient distributed optimization process for MUDA to process each dataset locally while We propose an efficient distributed optimization process for MUDA to process each dataset locally while encoding high-level learned knowledge in a shared latent embedding space." However, the novelty of the proposed method is questionable because a method for encoding high-level learned knowledge in a shared latent embedding space using GMMs has already been proposed for SFUDA [c, d, e, f, g], similar to the proposed method.

**Audience:**

Yes

**Claims And Evidence:**

The experiments, although somewhat limited in terms of the number of datasets, support the claims. The authors also included additional datasets in the revision.

---

> ### Author Response · Authors · 2022-10-14
> **Camera-Ready Version**
>
> Dear Action Editor,
>
> Thank you for managing the review process and the reviewers for their constructive comments that helped to prepare the camera-ready version. We have prepared and uploaded a camera-ready version of our manuscript according to your instruction. We have addressed the raised concerns as follows:
>
> 1.Regarding point (1), we added a discussion on comparing our results with non-private variants on page 9 of the paper, and extended the experiments from our discussion with reviewer Q4QX, and reported our findings in Table 3.
>
> 2.Regarding the first subpoint of (2), we checked the references and updated our claim. Please note [d,e] and both seem to utilize a trained source model for adaptation ([d] - Figure 1, [e] - Figure 1). We also note both these methods use a VAE-based architecture, which produces a sampling distribution at the latent feature level, similar to our method or other GMM-based methods from [c-g].
>
> 3.Regarding the second subpoint of (2), we discuss the works [c-g] Reviewer TfHu referenced in our Related Work section and provided a further discussion on privacy-preserving methods based on source domain approximations and how they compare to our approach in the Privacy in Domain Adaptation paragraph on page 3. We also compare recent SFUDA works and verify their Uniform MUDA performance in Table 5 to show the importance of our pooling strategy.
>
> 4.We have additionally included the reviewer's comments and results from discussions with the reviewers in either the main paper or as extensions to our Appendix.
>
> Best Regards,
>
> Our team